# Hard rock landforms generate 130 km ice shelf channels through water focusing in basal corrugations

Hafeez Jeofry[1,2], Neil Ross [3], Anne Le Brocq[4], Alastair G.C. Graham [4], Jilu Li[5], Prasad Gogineni[6], Mathieu Morlighem [7], Thomas Jordan[8,9] & Martin J. Siegert [1]

Satellite imagery reveals flowstripes on Foundation Ice Stream parallel to ice flow, and meandering features on the ice-shelf that cross-cut ice flow and are thought to be formed by water exiting a well-organised subglacial system. Here, ice-penetrating radar data show flow-parallel hard-bed landforms beneath the grounded ice, and channels incised upwards into the ice shelf beneath meandering surface channels. As the ice transitions to flotation, the ice shelf incorporates a corrugation resulting from the landforms. Radar reveals the presence of subglacial water alongside the landforms, indicating a well-organised drainage system in which water exits the ice sheet as a point source, mixes with cavity water and incises upwards into a corrugation peak, accentuating the corrugation downstream. Hard-bedded landforms influence both subglacial hydrology and ice-shelf structure and, as they are known to be widespread on formerly glaciated terrain, their influence on the ice-sheet-shelf transition could be more widespread than thought previously.

[1] Grantham Institute and Department of Earth Science and Engineering, Imperial College London, London SW7 2AZ, UK. [2] School of Marine Science and Environment, Universiti Malaysia Terengganu, Kuala Terengganu 21300 Terengganu, Malaysia. [3] School of Geography, Politics and Sociology, Newcastle University, Newcastle upon Tyne NE1 7RU, UK. [4] Geography, College of Life and Environmental Sciences, University of Exeter, Exeter EX4 4RJ, UK. [5] Center for the Remote Sensing of Ice Sheets, University of Kansas, Lawrence 66045 Kansas, USA. [6] ECE and AEM Departments, The University of Alabama, Tuscaloosa, AL 35487, USA. [7] Department of Earth System Science, University of California, Irvine 92697 CA, USA. [8] Department of Geophysics, Stanford University, Stanford 94305 CA, USA. [9] School of Geographical Sciences, University of Bristol, Bristol BS8 1SS, UK. Correspondence and requests for materials should be addressed to M.J.S. (email: m.siegert@imperial.ac.uk)

From the grounding line of several ice streams, meandering surface channels in the adjacent ice shelves have been observed in moderate-resolution imaging spectroradiometer (MODIS) ice-surface imagery, and linked to upward-incised channels at the ice-shelf base; their cause being the surface elevation differences of buoyant thick versus thin ice[1–3]. In this paper, we refer to these surface channels as M-channels (Fig. 1). MODIS imagery also reveals lineations in the surface of the ice sheet that are orientated parallel to ice flow; these are flow-stripes formed by ice-flow processes, often during lateral convergence[4].

It has previously been demonstrated that M-channels correspond with the likely exit point of subglacial water at the mouths of ice streams that are inferred to have flat beds[1], or sedimentary landforms (eskers) that route water[3]. Because of this, M-channels are thought to be evidence of a well-organised subglacial-hydrological system, channelized by upward melting into the grounded ice by the basal water. As it exits the grounded ice the subglacial water forms a buoyant plume due to being fresher than water within the ice shelf cavity. It then entrains the warmer cavity

water, and melts the underside of the floating ice to form an upward-incised channel[5], which we refer to as U-channels. Hence, M-channels and U-channels are co-existent. Similar findings have been reported across a variety of ice-stream grounding lines across Antarctica[2]. The flow of ocean water within the cavity has also been shown to lead to upward-incised ice-shelf channels in, for example, the floating ice at the margin of Pine Island Glacier[6].

While surface ice-shelf melting does not cause M-channels, if such melting occurred the water produced would be preferentially routed into and along M-channels due to the linear depression in the surface topography, potentially melting them downwards[7]. Given this, and the importance of maintaining ice shelf integrity in support of ice-sheet stability in the context of atmospheric warming[8,9], it is necessary to identify and understand the hydrological supply of water responsible for the sizable selective upward linear erosion observed, particularly in deep marine settings in West Antarctica.

Here we inspect ice-penetrating radar data across the Foundation ice stream (FIS) and Filchner-Ronne Ice Shelf (FRIS) to

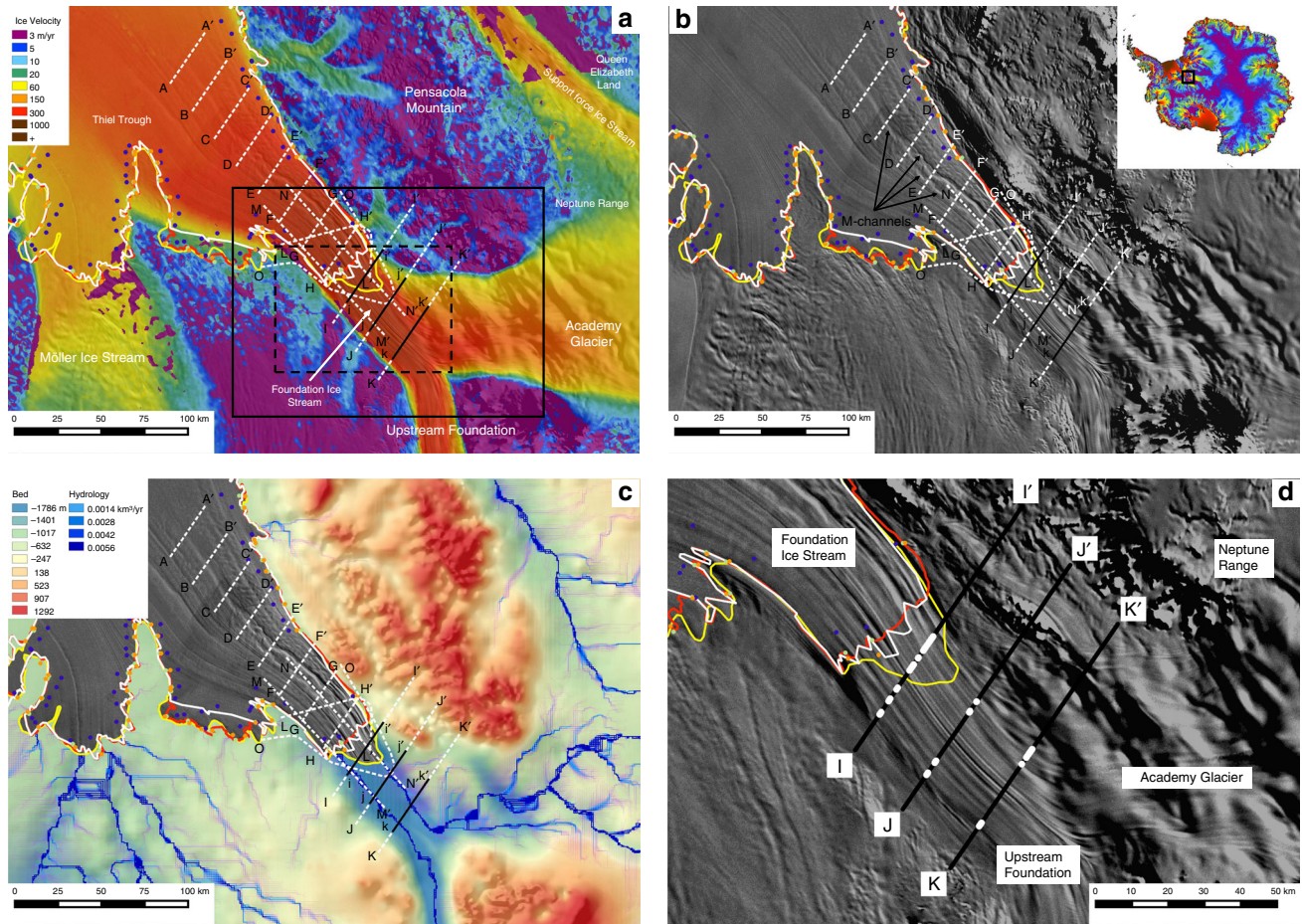

**Fig. 1** Ice-surface imagery, bed elevation, ice-surface velocity, and subglacial water flow for the Foundation ice stream. **a** Ice-surface velocities[49] underlain by MODIS ice-surface imagery (solid-line box outlines refer to magnified region in (**d**) and dashed-line box is in Supplementary Figure 4). Grounding points from the Ice, Cloud, and land Elevation Satellite (ICESat) laser altimetry are as follows: in blue—hydrostatic point; orange—ice flexure landward limit; and green—break in slope[47]. Grounding lines are from the Antarctic surface accumulation and ice discharge (ASAID) (red)[13], the differential satellite synthetic aperture radar interferometry (DInSAR) (yellow)[14], and the mosaic of Antarctica (MOA) (white)[48]. **b** MODIS ice-surface imagery of the Foundation ice stream trunk. Meandering lineations downstream of the grounding line are noted as "M-channels". Radar transects, annotated as in other figures, are shown. The inset denotes the location of the study region in Antarctica. **c** Bed elevation[23] and subglacial-hydrological pathways (calculated as discharge rates). **d** Regions of enhanced basal reflectivity (>5 dB, noted in white lines) along three transects, superimposed over MODIS imagery[50] and the grounding lines as in (**a**)

link ice-surface lineations observed in MODIS imagery to basal features in the ice sheet and ice shelf. Our aim is to better understand processes at the transition between grounded and floating ice, and the role that subglacial conditions can have in determining ice-shelf morphology and, potentially, structural integrity. We reveal mega-scale hard-bedded subglacial landforms at the grounding zone of FIS, which cause the ice base to become corrugated as it starts to float. The basal landforms also modulate the flow of basal water, feeding it into a corrugation peak which then develops the corrugation for more than 130 km from the grounding zone. Similar hard-rock basal landforms are known to exist across formerly glaciated terrain in many regions of Antarctica, and landscapes beneath former northern hemisphere ice sheets, suggesting their influence on ice dynamics may be more significant than appreciated both in terms of contemporary ice dynamics and past ice-sheet behaviour.

## Results

**Foundation ice stream**. Despite being a major Antarctic ice stream, the subglacial topography and basal environment of the FIS are poorly characterised[10,11]. The FIS has its trunk in West Antarctica and a wide complex drainage basin with about half its ice supplied by the Academy Glacier and Support Force ice stream in East Antarctica and the other half from West Antarctica. Location of the grounding line of FIS, which is around 2 km below sea level, is ambiguous depending on whether surface slope or tidal flexure is used[12-14]. The separation between two of the proposed grounding lines is up to 18.5 km, indicating a possible complex transition between grounded and floating ice (Fig. 1).

The complex nature of the FIS catchment will likely reflect an equally complex glacial history[15], especially considering ice-dynamic changes that have occurred in West Antarctica over the last few thousand years[16,17], and around and upstream of South Pole over longer time periods[18,19]. Numerous "active" subglacial lakes—those that experience outbursts or infilling of water due to periodic change in their hydropotential gradients[20,21]—exist beneath both Support Force and Academy Glaciers[22], indicating that Foundation Ice Stream is fed with significant volumes of basal water, particularly of East Antarctic origin (Supplementary Figure 1).

The main trunk of FIS, and its grounding line with the FRIS, lies ~2 km below sea level. The bed beneath the ice stream trunk occupies a deep U-shaped fjord[23]. While the grounding line is not located on or near a major reverse bed slope, the depth of the grounding line is particularly interesting from the perspective of ice-sheet processes and stability in deep-marine settings, as grounding line retreat in other locations may lead to similar situations in the future (e.g., Thwaites Glacier in West Antarctica, and Lambert and Totten Glaciers in East Antarctica).

Satellite imagery shows the trunk of FIS to be characterised by a series of flowstripes[4,24] (Fig. 1a). Downstream of the grounding zone, M-channels are clearly demonstrable in MODIS imagery for a distance of ~130 km. One M-channel abuts the grounding line derived from surface slopes, and appears to extend in reduced amplitude a further 10 km toward the grounded ice stream. A second M-channel occurs about 20 km from the same grounding line. These two M-channels merge ~100 km from the grounding zone, and this combined channel fades out ~30 km downstream (Fig. 1a).

**Radar bed-topography, grounding line position, and subglacial landforms**. Airborne ice-penetrating radar is the key technique to measure the morphology and condition of beds of large ice sheets. The radar data used in this study were compiled from two main sources: flights conducted by the Center for Remote Sensing of

Ice Sheets (CReSIS) as a part of the NASA Operation IceBridge (OIB) mission in 2012, 2014, and 2016; and a survey of the Institute and Möller ice streams undertaken in 2010/2011 (IMAFI)[25]. Specifically, we analyse twelve equally spaced transects aligned orthogonal to ice flow (A–A' to K–K'), two near the grounding zone (L–L' and O–O'), and two others parallel to the axis of FIS (M–M' and N–N') (Figs. 1 and 2, and Supplementary Figure 2). Flow-orthogonal radar data are particularly useful for delineating the lateral extent and heights of flow-parallel bedforms, whereas flow parallel transects can be used to measure bed roughness along flow, placing into context the flow-orthogonal roughness[26]. Radar data were used to form a digital elevation model of the region[23], which updates significantly the Bedmap2 version[27]. The new DEM also benefits from an interpolation procedure that accounts for ice-flow mass conservation, as demonstrated primarily in Greenland[28]. Using this new DEM, and surface elevations from Cryosat2, we calculate subglacial water flowpaths[29] (Fig. 1c).

The two radar transects aligned parallel to the FIS axis reveal the ice stream bed to be extremely smooth (Fig. 2; Profile M–M' and Profile N–N'). These data also reveal classic ice-surface and ice-base profiles as the grounded ice sheet transitions to a floating ice shelf. In M–M', the grounding line matches that derived from both ice-surface slopes[12] and tidal flexure[14]. However, in N–N', the proposed grounding lines are separated by ~20 km (Fig. 2). A good determinant of deep-water (and thus where the ice sheet transitions to the ice shelf) can be derived from radar scattering properties of the basal interface. The so-called "abruptness" of the echo waveform (a parameter where higher values are associated with specular reflections and lower values are associated with diffuse scattering)[30] across N–N' reveals a step-wise change in basal water depth upstream of the "surface slopes" grounding line, and downstream of the "ice flexure" line (Fig. 2, Supplementary Note 1, Supplementary Figure 3). Using a classic ice-flotation equation[25] a 5 m increase in sea level, which is in line with the tidal range in this region[31], is able to shift the grounding line upstream by at least ~10 km (Fig. 2). Hence, the discrepancy between the grounding lines may be resolved by tidal-forced uplift of a substantial portion of the FIS trunk.

Orthogonal to ice flow in the trunk of FIS, three radar transects show significant bed roughness due to massive flow-parallel ridges (the peaks of the ridges align well with flowstripes; Supplementary Figure 4), which are divided into two distinct sets related to the origin of the ice (i.e., one set from upstream FIS and another from Academy Glacier). The ridges across the Academy Glacier side of the trunk are of the order of 100–300 m high and 2–3 km wide (Fig. 2; Profile I–K; Supplementary Figure 5). We are able to track these bed ridges in cross-flow radar transects along the trunk of the FIS (transects J–J' and K–K', Figs. 1 and 2, and Supplementary Figure 6) ~40 km inland of the grounding zone. The bed-ridges are similar to those seen in other regions of the ice base[32], and over formerly glaciated terrain and offshore marine regions where enhanced flow has once occurred[33-39] (Fig. 3). The scale of the bedforms, and their relation to hard-bedded features seen elsewhere, suggests they are formed predominantly of hard rock. We do not believe they can be composed of sediment, as most relict streamlined large-scale sedimentary lineations mapped on the Antarctic continental shelf have amplitudes one (and often two) order(s) of magnitude less than those imaged across the trunk of FIS[40]. In the few areas where sedimentary bedforms forming part of an active dilatant till layer have been imaged by radar, the largest is only ~20 m high[32]. We do not believe the FIS bedforms are 'eskers', as has been observed elsewhere in Antarctica[3], as besides the size consideration we do not observe any sinuosity (although we may undersample the morphology). Furthermore, the amplitude of the

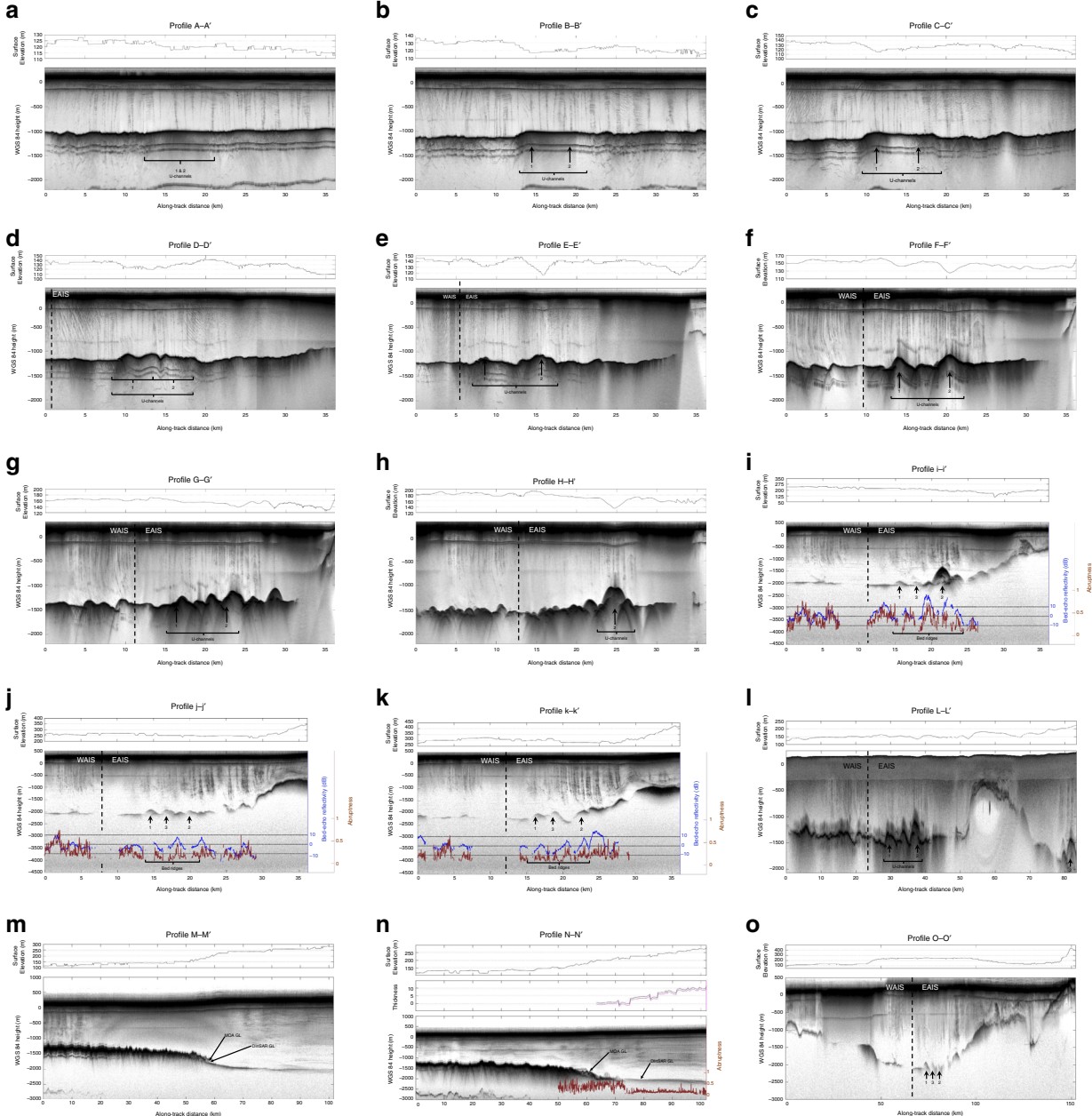

**Fig. 2** Radar transects revealing subglacial landforms in grounded ice and basal channels incised upwards beneath floating ice. **a** A–A', **b** B–B', **d** C–C', **d** D–D', **e** E–E', **f** F–F', **g** G–G', **h** H–H', **i** i–i', **j** j–j', **k** k–k', **l** L–L', **m** M–M', **n** N–N', and **o** O–O', as located in Fig. 1. Extended versions of i-i' (I-I') j-j' (J-J') and k-k' (K-K') can be found in Supplementary Figure 2. Also for these three transects, basal reflectivity (relative values, zero mean) and the echo abruptness are provided. For transect N–N', the echo abruptness is provided (revealing a step change between grounded and floating ice) with the thickness of ice above the level of flotation (with a tidal range of 0 m, black line; and +5 m, pink line). The identification of three flow-parallel bedforms and U-channels, described in Supplementary Figure 5, are also shown (see Supplementary Figure 6 for locations). Separation between ice sourced from East versus West Antarctica is noted by the dashed black line. Surface elevation profiles (from the aircraft altimeter) are also shown

bedforms remains largely unchanged upstream of the grounding line in contrast to 'tadpole' shaped eskers proposed elsewhere that have higher amplitudes only near the grounding line[3]. It is interesting to also note that, to our knowledge, no eskers have been identified across the formerly glaciated Antarctic continental shelf. This strongly suggests that the FIS bedforms are erosional landforms with a lithified core[40]. Any actively deforming till layer may be superficial at the scale of the bedforms, though may likely be present between them (as is evident in the very flat bed separating bedforms). The cross-sectional dimensions of the FIS

bedforms match well to large streamlined linear ridges that have been mapped across a number of formerly glaciated regions in Antarctica, where they are carved in hard crystalline bedrock[41], and which are plastered by a till sheet at least several metres thick (Fig. 3). We presume that ice flow is organised around the bedforms to maximise flow efficiency, with their streamlining the result of the relatively stable flow regime of FIS afforded by steep, constraining lateral topography. Large-scale hard-bed bedforms are commonly associated with convergent flow of ice[33], and so their presence in the trunk of Foundation ice stream, fed by two

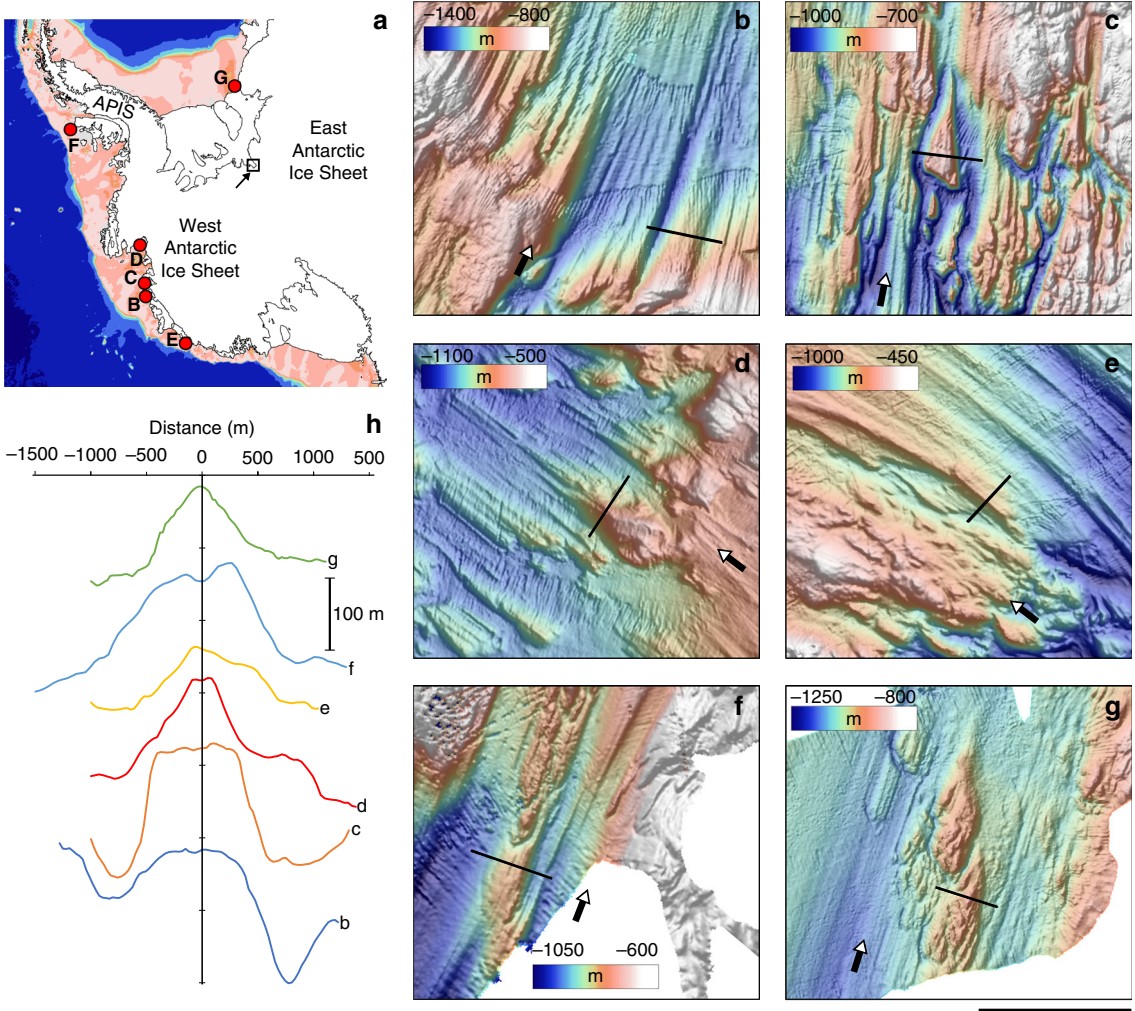

**Fig. 3** High-resolution multibeam echo-sounder swath bathymetry datasets for the Antarctic shelf, showing examples of relict hard-bedded landforms with dimensions directly comparable to those imaged with radar beneath the trunk of Foundation ice stream. **a** Location map of six individual sites spanning the East Antarctic, West Antarctic, and Antarctic Peninsula coastlines; letters refer to the data shown in the subsequent panels, **b–g**. **b**, **c** Subglacial bedforms carved into crystalline basement in front of the Getz B ice shelf, and Dotson Ice Shelf, respectively[41,51]. Grid cell size 30 m. **d** Crag-and-tail like bedforms in a crystalline substrate seaward of Pine Island Glacier[33]. Grid cell size 35 m. **e** Inner shelf bathymetry from the Hobbs Coast region of the westernmost Amundsen Sea/eastern Ross Sea[52]. Grid cell size 30 m. **f** Hard-bedded landforms on the middle continental shelf north of the Wilkins Ice Shelf. Data collected on cruise JR179[51]. Grid cell size 30 m. **g** Sea-floor landforms in bedrock in the southeastern Filchner Trough. Data collected on cruise JR244[53]. Grid cell size 25 m. **h** Cross-profiles of landforms as shown in panels **b–g** (thin black lines). Together panels **b–h** demonstrate that mega-scale hard-bedded landforms are common around the Antarctic coastline. In all cases, the larger bedforms are overprinted or found adjacent to smaller-scale landforms with reduced amplitudes and wavelengths that are more typical of sedimentary and bedrock-moulded sub-ice stream landscapes. We presume that these subglacial bedforms also have an effect in corrugating the ice shelf base but to a much lesser extent than the rarer and more prominent hard-bedded glacial landforms. Thick black scale bar is 5 km. Filled arrows show former ice flow direction. APIS: Antarctic Peninsula Ice Sheet

distinct and sizeable tributaries, is a reasonable supposition. We are unable to determine the precise nature of the bedforms, however, (i.e., whether they are rock drumlins or roche-moutonnées). Between the bedforms, which are separated by less than 1 km in almost all cases, we observe the bed to be very flat, indicative of weak water-saturated sediments. While subglacial morphology on the WAIS-sourced side of the FIS trunk appears more subdued than that of the EAIS-sourced side, a similar glacial geological morphology is apparent with small bedrock highs (up to ~100 m high) interspersed by flat, presumably sediment-draped, surfaces.

**M-channels, U-channels, and basal water flow**. Across the ice-shelf base, several flow-orthogonal radar transects reveal a series

of U-channels (Fig. 2; Profiles A–A' to L–L'), which map directly beneath M-channels confirming their association noted previously[1–3]. Because the ice is afloat, the surface elevation above a U-channel (i.e., the elevation of the M-channel) will be less than the surrounding ice shelf surface. Hence, we are able to confirm U-channels from this additional diagnostic (Fig. 2).

In one location, within the FIS grounding zone, a distinct reflection ~800 m in height has been measured (Fig. 2; Profiles i-i' and L–L'). Close inspection of the radargram reveals the reflection to involve multiple "peaks", suggesting that one or more reflections may be from offline reflections. As transect i-i' is orthogonal to flow, the 800 m peak reflection is likely to be sourced downstream. We believe this reflection is from a U-channel, rather than a rock pinnacle (especially given the high

reflectivity of the peak; Supplementary Note 2, Supplementary Figure 7). If the U-channel was positioned vertically beneath the aircraft we would expect a depression in ice-surface elevation. However, the transect altimetry reveals a slight increase in surface elevation. Hence, we believe transect i-i' lies very close to the transition between the bedforms and U-channel initiation. Downstream of the grounding line, and along the same line of ice flow, a similar basal reflection is observed (at around 400 m in height) (Fig. 2 H–H'). This time, however, a depression is observed in the ice surface (i.e., coincident with an M-channel).

The M- and U-channels originate from the Academy Glacier side of the FIS trunk. This is consistent with subglacial water flow-routing calculations (Fig. 1, Supplementary Figure 1), which demonstrate three important features about the hydrological system. First, the majority of the water from the Academy catchment is supplied to the head of the main FIS trunk, and then flows parallel to ice flow and, hence, the bedforms. Second, the FIS captures two extremely large catchments of subglacial water; one emanating from the interior of West Antarctica, and one from the South Pole region in East Antarctica. In the latter catchment, water is preferentially routed into Academy Glacier to the detriment of Support Force Glacier. The third aspect is that the FIS hydrology comprises two distinct components, derived from the two catchments, with no interaction between them. Hence, one side of the FIS trunk is supplied by water from West Antarctica, while water in the other half comes from East Antarctica.

The main drainage pathway from East Antarctica exits the FIS trunk in the zone where there is grounding line uncertainty and the largest (~800 m in height) U-channel (labelled U-2 in Fig. 2, Supplementary Figure 4). The channel decreases in height, but not in elevation or width, down-ice flow as noted in earlier studies[1], due to mixing of the waters and freezing onto the ice-shelf base. Importantly, the width of the U-channel is very similar to the width of the bedforms upstream, further indicating their association. Another U-channel (U-1 in Fig. 2, Supplementary Figure 5), which starts ~20 km from the grounding line derived from surface slopes (see Fig. 2, profile G–G'), is also of similar width to the bedforms. However, we do not observe its existence at the grounding line (see Fig. 2, profile H–H'). The two U-channels merge ~100 km from the grounding zone (Figs. 1, and 2 profile C-C'), and by ~130 km the combined U-channel has largely disappeared.

Based on the correspondence between surface and basal features with the modelled hydrological flow paths below the grounded ice, we conclude that U-2 at the grounding zone is formed by the action of a well-organised basal hydrological system ejecting an outflow of water to the ice-shelf cavity. This water mixes with warm cavity ocean water to form a plume that is less saline than cavity water and warm enough to melt the ice above. U-1 is less obviously consistent with formation by subglacial water. Instead, tidal pumping of water may keep channel U-1 open. Because the widths of both U-channels are so similar to the bedforms, and since we are able to connect one of them to a specific bedform, we believe U-1 may have been created in a similar way to U2. If this is the case, the gap between U-1 and the grounding zone may reflect grounding line migration and/or past hydrological variability.

**Radar reflectivity measurements**. Precisely how the water flows across the ice sheet bed within the trunk of FIS requires analysis of the reflectivity and scattering properties of the radar bed-echo data. The flow-parallel, likely hard-bedded, subglacial landforms influence the flow direction of water (especially as the surface

slope is very low[42]). Basal power was extracted from the radar data[43] along profiles i-i', j-j' and k-k'. Relative basal reflectivity values (used to discriminate basal water) were then obtained performing a separate attenuation correction for radar traces in East and West Antarctica (see Supplementary Note 1 and Supplementary Figure 3 for more details).

The radar reflectivity characteristics of the FIS bed are consistent with significant well-organised flow of water within FIS (Fig. 2). The overall dB range for basal reflectivity is ~30 dB (refer to Supplementary Note 1 for the frequency distribution). This is consistent with the predicted range for subglacial materials at radar frequencies[44,45], although it is important to bear in mind that the FIS is a region where anomalous radar power losses (e.g., due to crevassing) are likely to be present. The bed of the trunk of the FIS is likely to be above pressure melting point and comprised (in between the hard-bedded landforms) of subglacial till. In this scenario, the basal reflectivity hierarchy is likely to correspond to basal water greater than ~5 m deep (highest values), saturated till, hard-bedded regions, and dry till (lowest values)[44,45]. The oscillations in basal reflectivity along transects i-i', j-j' and k-k' are therefore consistent with material transitions from drier surrounding regions to water channels (the reflectivity peaks). Supporting evidence for the water channel locations comes from the correlation between the reflectivity peaks and the abruptness peaks. Specifically, high values of both are consistent with water being present[43].

While the subglacial flow routing model shows the path of basal water is consistent with flow alongside the bedforms in the trunk of the ice stream, we regard the most compelling evidence of water at the bed to come from the radar data. Within the region of grounding zone uncertainty, where the highest (~800 m) U-channel is measured, reflectivity is above +10 dB in three regions; one over a flat region of the bed, and two associated with the fringes of the U-channel (Profile i-i'). The U-channel in profile i-i' (Fig. 2) is an offline reflection, and has a reflectivity greater than the bed between the landforms, similar to that observed over deep-water subglacial lakes (Supplementary Note 2).

Upstream, across the permanently grounded ice-stream trunk, points where reflectivities reach ~+10 dB are observed (Profile j-j'). Further upstream, an additional three points with reflectivities ~+10 dB are seen, and in one location on the fringe of the trunk reflectivities are >+10 dB (Profile K–K'). Basal reflectivities above 5 dB are mapped in Fig. 1. They plot along a discrete flowstripe, confirming the flow of water to be well-organised and aligned with flow and the bedforms. We see no evidence for basal water cross-cutting the ice-stream flowline in the trunk of the ice stream (i.e. they are channelled by the basal landforms).

## Discussion

Because of the similar widths, and spatial relationship, between the subglacial landforms and the sub ice-shelf U-channels (Supplementary Figure 5) we propose that the bedforms are dictating the position and form of the U-channels. Since the ice will mould around the subglacial bedforms[46], as the ice-sheet becomes afloat the base of the ice sheet will inherit a "corrugated" morphology from the bedforms. The bedforms thus modulate the upstream flow of subglacial water, allowing a focused stream of water to flow upwards into the peak of a U-channel corrugation across the grounding zone (Supplementary Figure 8). The combination of subglacial hydrology and hard-rock bedforms across and upstream of the grounding zone is therefore critical to the propagation of U-channels downstream. This finding, though not previously reported, is unlikely to be unique in Antarctica. For

example, the Institute ice stream, which lies above a predominantly flat bed, indicative of wet sediment, is also associated with a small region of rough bed across the deepest parts of the Robin Subglacial Basin coinciding with a U-channel[1], and could be explained by "glacial excavation" of sediment revealing hard-rock landforms[21]. As a consequence, the association between U-channels and hard-rock flow-parallel landforms could be generally applicable across the margins of marine ice sheets both now and in the past.

While our observations of subglacial landforms and basal water dictating U-channel genesis and development are clear, the reason for why U-channels can exist so far from the grounding line, and thus from the source of subglacial water, remains less certain. One explanation is that tidal pumping of water can maintain the channel once developed. For FIS, the tidal range is at least 5 m, meaning there is huge potential flux of water into and out of the ice-shelf cavity daily. This may provide an explanation for why some of the U-channels meander and merge downstream of the grounding zone, if they are actively being reworked by tidewater.

Another unresolved issue relates to the significance of subglacial water in creating the U-channel at the grounding zone. If basal water is not present, then we may expect the channel to experience creep closure from the surrounding ice as it flows around the bedform and the U-channel would fail to form. Further, if the supply of basal water was switched off to an existing U-channel it may cease to initiate. This may explain why U-1 is not observed at the grounding zone; instead it is both advecting downstream and being held open by tidewater. If this idea is correct, U-1 may be evidence of temporal variability in the supply of basal water from FIS. All Antarctic ice shelves are potentially sensitive to atmospheric warming induced surface melting, but those with weaknesses inherited from upstream glaci-geological and subglacial-hydrological processes may be particularly vulnerable compared with ice shelves fed by ice streams on flat beds.

## Methods

**Radar data**. The radar data used in this study were compiled from two main sources. First, a survey of the Institute and Möller ice streams undertaken by the British Antarctic Survey (BAS) in 2010–2011. The survey used the BAS Polarimetric radar Airborne Science Instrument (PASIN), operating at a centre frequency of 150 MHz, with a 10 MHz bandwidth and a pulse-coded waveform acquisition rate of 312.5 Hz. Second, a series of geophysical flights conducted by the CReSIS during the NASA Operation Ice Bridge programme in 2012, 2014, and 2016, which used the Multichannel Coherent Radar Depth Sounder system developed at the University of Kansas. The system operated with a carrier frequency of 195 MHz and a bandwidth of 10 MHz in 2012 and 50 MHZ in 2014 onward.

**Radar data processing—PASIN**. Chirp compression was applied to the along track data. Unfocused synthetic aperture (SAR) processing was used by applying a moving average of 33 data points, whereas two-dimensional SAR (i.e., focused) processing based on the Omega-K algorithm was used to enhance both along track resolution and echo signal noise. Doppler filtering was used to remove the backscattering hyperbola in the along track direction. The bed echo was depicted in a semi-automatic manner using ProMAX seismic processing software.

**Radar data processing—CReSIS**. The data were processed in three steps to improve the signal-to-noise ratio and increase the along-track resolution. The raw data were first converted from a digital quantization level to a receiver voltage level. The surface was captured using the low-gain data, microwave radar or laser altimeter. A normalized matched filter with frequency-domain windowing was then used for pulse compression. Two-dimensional SAR processing was used after conditioning the data, which is based on the frequency wave number (F–K) algorithm. The F–K SAR processing requires straight and uniformly sampled data, however, which in the strictest sense are not usually met in the raw data since the aircraft's speed is not consistent and its trajectory is not straight. The raw data were thus spatially resampled along track using a sinc kernel to approximate a uniformly sampled dataset. The vertical deviation in aircraft trajectory from the horizontal flight path was compensated for in the frequency domain with a time-delay phase

shift. The phase shift was later removed for array processing as it is able to account for the nonuniform sampling; the purpose is to maintain the original geometry for the array processing. Array processing was performed in the cross-track flight path to reduce surface clutter as well as to improve the signal-to-noise ratio. Both the delay-and-sum and minimum variance distortionless response (MVDR) beamformers were used to combine the multichannel data, and for regions with significant surface clutter the MVDR beamformer could effectively minimize the clutter power and pass the desired signal with optimum weights.

**Radar data measurements**. In both datasets, the waveform was retrieved and sequenced according to its respective transmit pulse type. The modified data were then collated using MATLAB data binary files. A nominal value of 10 m is used to correct for the firn layer during the processing of ice thickness, which introduces an error of the order of ~3 m across the survey field. This is small relative to the total error budget of the order of ~1%. Finally, the GPS and RES data were combined to determine the ice thickness, ice-surface and bed elevation datasets. Elevations are measured with reference to WGS 84. The ice-surface elevation was calculated by subtracting terrain clearance from the height of the aircraft, whereas the bed elevation was computed by subtracting the ice thickness from the ice-surface elevation.

## Data availability

Airborne radar data used in this study are freely available at the CReSIS website. The digital elevation model of the Foundation ice stream, and radar data used to build it, are available at [https://doi.org/10.5194/essd-10-711-2018]. In addition, all relevant data are available from the corresponding author.

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

## Acknowledgements

CReSIS radar data were collected as a part of NASA grant # NNX10AT68G and a significant resources for processing these data were provided through ANT # NT-0424589, and with additional support from the University of Kansas. IMAFI radar data were collected through UK NERC AFI grant NE/G013071/1 to MJS. We thank Quantarctica and the Norwegian Polar Institute for datasets and colour schemes used in Figs. 1 and 3 (see: http://quantarctica.npolar.no/about.html). Supplementary Figure 4 uses the Bedmap2 depiction of the Antarctic subglacial topography (https://www.bas.ac.uk/project/bedmap-2/). We would like to acknowledge the hard work done by the Moderate-Resolution Imaging Spectroradiometer (MODIS) Mosaic of Antarctica (MOA) team in compiling the MOA imagery used in Fig. 1, and Supplementary Figures 1 and 4 (https://nsidc.org/data/moa). We would also like to thank Dustin Schroeder at Stanford University for his helpful comments on the reflectivity analysis.

## Author contributions

H.J. processed and analysed the radar and satellite data, and drew the figures, under supervision from M.J.S. and N.R. A.L.B. advised on the subglacial water flow model. N.R. acquired the IMAFI data and contributed widely to the paper. J.L. and P.G. contributed

the CReSIS data and helped with data processing. M.M. provided the refined bed topography. A.G. advised on hard-bedded morphological analogies, and provided Fig. 3. T.J. carried out the radar reflectivity analysis. M.J.S. wrote the paper with input from all co-authors.

## Additional information

**Competing interests:** The authors declare no competing interests.

