## [Peer Review File · Nature Communications]

Reviewers' comments:

Reviewer #1 (Remarks to the Author):

Summary

Jeofry et al. present new observations from radar data covering the Foundation Ice Stream in West Antarctica. They use modelling of the subglacial hydrology in conjunction with interpretations of the basal reflectivity to draw their conclusions. The latter can be summarized as follows:

1. Ice-shelf channels (curvilinear tracks of thin ice which frequently occur on Antarctic ice shelves) in the ice shelf connected to the FIS are induced by topographic undulations located upstream of the grounding line.
2. The topographic undulations are flow-parallel, hard-bed landforms (2—3 km wide; 100-200 m high; 10s of kilometers long; spacing 1 km; potentially covered by till). Implicitly, it is concluded that the hard-bed landforms cause formation of flowstripes at the surface.
3. The flow-parallel, hard-bed landforms determine the location of water-filled, subglacial channels which terminate (at the grounding line) exactly where ice-shelf channels are observed. The U channels are thus further incised upwards through localized, plume-driven melting.
4. The joint observations also explain the meandering/merging of ice-shelf channels farther downstream.
5. The grounding line of FIS is lifted up on tidal cycles explaining the differences of 18.5 km in published grounding-line datasets.

This paper tackles an important point in glaciology linking to the stability of Antarctic ice shelves and their role in buttressing the upstream ice flux. There is currently no consensus if ice-shelf channels stabilize or destabilize Antarctic ice shelves, and this is partially due to their unknown origin. This paper has potential to investigate this matter further.

General Remarks

The findings confirm some aspects (i.e. conclusions 1 & 3) in a study that we published earlier (Drews et al., Nat. Com. 2017) but also provide alternative explanations (i.e. conclusions 2 & 4). The latter may apply to our previous study (as alluded in l. 243), but could also describe a different phenomenon. Although I provide much criticism in the comments below, I want to stress that I am not defensive against the alternative explanations. All comments are fully geared to make this paper more solid and also more easy to follow. Unfortunately, I find the paper currently imprecise in many places. Arguments are not consistently applied, and not enough information is at hand to check their validity. I have some confidence in conclusions 1 and 2, but am unconvinced by conclusion 3-5. Therefore, I don't think that the paper can be published in its current form. Below I have some questions/remarks which I hope the authors will find useful for improving the current version.

Specific Remarks

Subglacial water routing and its relation to the basal landforms:

The subglacial hydrology routes based on the updated thickness model show now three instead of one water flowpath previously. As I understand it right now, the authors only updated the ice thickness for their novel calculations, but not the surface elevation of Bedmap 2 (which is outdated by now). The following questions come to mind:

- Wright et al. (Geophys. Res. Lett., 2008) show that the surface topography is very important for calculating the subglacial drainage network. Is this not the case here and why not?
- What resolution does the surface DEM have?
- Do the flowstripes have a correspondence in the surface topography, and if so, would that impact the prediction of the subglacial drainage routes?
- How different are the predicted flowpaths when using a gridded vs. a mass-conserving bed? This is important, because I suspect that the gridded version does not make an along-flow connection between the subglacial landforms observed in the radar transects J-J', K-K'. However, I presume that such a connection is required if the basal topography should guide the subglacial water flow. I don't know what gridding + mass conservation changes in that respect. Outsourcing these points to the "Jeofry et al., in press" reference only works if those points are discussed in detail there.

I understand that those are quite detailed questions, but at the same time, the authors draw detailed conclusion from the analysis (i.e. the predicted flowpaths are reliable on 1 km scales, which is the scales of the landforms/M/U channels). Without a sensitivity analysis of the subglacial water routing as a function of the input variables, I don't see how this point can be justified. (I apologize if I missed essentials of the subglacial hydrological modelling, of which I am not a specialist).

Moreover, I currently don't see improvement to the results which have already been published by LeBrocq et al., Nat. Geosc., 2013. The suggested improvement is that the subglacial landforms will "determine the position of any (sic!) buoyant subglacial water emerging from the grounding zone." (l. 234). Yet, "Only the smaller 'East Antarctic' basal water route is aligned with flow-parallel bedforms." (l. 175). Those are directly conflicting statements. (Also, if the "East Antarctic" route is aligned with multiple landforms (plural), does it mean it crosses the landforms 1-4?)

None of the Figures convey the link between the subglacial water routes and the landform sufficiently (e.g. In Figure 2b the exit points at the grounding line are overprinted by the radar profile lines and the subglacial landforms are not marked, nor are the relevant flowstripes (there are many).). Also the M-channels are only marked at one location, but the underlying image does not allow to trace the flow stripes back to the grounding line. As a last point, the radar reflectivity analysis in the transects J-J' and K-K' should be linked to the projected water flow paths of Figure 1b, not not the hydrological potential. I don't see how people are suspected to infer the most likely water flow paths from the transects of the hydrological potential and how these related to the radar reflectivity values.

Given all the points I mentioned above, I hope you understand my reservations about how these findings are portrayed.

Further incision of ice-shelf channels by buoyant melt water plume: In multiple places, the authors suggest that U channels are further incised by melt-water plumes seawards of the grounding line. There is already published evidence for this (e.g. LeBrocq et al. 2013; Marsh et al., GRL, 2015).

- What is the additional value of this study in this regard?

If plume melting was very relevant, wouldn't you expect growing of the U-channels seawards? Instead, "U-channels remain relatively consistent across the grounding line and down flow" (l. 157). How does this fit together? From my point of view, you could argue that ice-shelf channels would close without plume melting due to lateral inflow (Drews, TC, 2015). I apologize for advertising my own reference here, but something needs to be done to strengthen the inference about plume melting and its relation to the topographically induced perturbations from farther upstream. Otherwise I don't see the additional value compared to the previous findings.

The meandering of M/U-channels: The abstract states that this analysis "explains the origin of meandering surface channels" (l. 31). I don't see progress in this paper explaining the meandering of the channels on the floating ice shelves. In fact, l. 178 explicitly states that the authors cannot distinguish whether the merging of surface channels has occurred on the ice shelf or at the grounding line.

- So how is the meandering then explained?

The location of the grounding-line: The authors suggest that the grounding line is lifted up during high-tide over an along-flow section of at least 18.5 km. This supposedly explains the differences between the DInSAR grounding-lines and other grounding-lines mapping the break in surface slope (l. 117). I am not aware of any studies that have shown that tidal uplift occurs over such long distances, and this should be put into context with other studies (e.g. K. Christianson et al. in publications of the Whillans Ice Stream; or Konrad et al., Nature, 2017).

- However, the biggest problem for this paper is that it remains unclear if the profile I-I' then images a floating or a grounded section of the ice sheet. If it is the former, the radar reflectivity analysis of this transect should detect basal water everywhere. However, it doesn't, so I am left with the expression that it is grounded. Does that mean the profile was collected during low-tide?
- Line 154, on the other hand, describes the feature in the I-I' transect as an 800 m U-channel (and not as a flow-parallel landform), this makes me think again that you consider the U-channel to be water filled (i.e. floating?). What does this then mean for the analysis of basal reflectivity which does not detect basal water everywhere?
- I also disagree with the statement that there is no reason to doubt the DInSAR grounding line (l. 116). You can only say this, if you know which interferograms were used to detect the grounding line at this location. Some of them were not double-differenced, and even if they were, it is not always trivial to detect the landward limit (it depends on the differential tidal uplift between the image acquisition, and this can be zero in some cases). The authors must provide much more evidence to justify their

statement about an >18.5 km grounding line migration by tides, only relying on previously published datasets is not enough.

- One of the most straightforward ways to do this would be to attempt to locate the grounding line in the available radar data using the basal reflectivity of the along-flow transect N-N'. This seems much easier than to detect localized melt water channels in the across-flow transect, and I wonder why this was not done.

Relation of flowstripes and basal landforms: The basal landforms detected in K-K' and J-J' are linked using the flowstripes. This is ok if (but only if), the flowstripes are caused by the bedforms. Glasser & Gudmundsson (T. Cryosph., 2012) and many other publications, however, also propose other mechanisms for the formation of flowstripes. The most prominent to consider here is the formation of flowstripes due to glacier confluence which is clearly the case here.

- How can you exclude other mechanisms than bed bumps for the formation of flowstripes? If this can't be done, then linking the bedforms between the two radar transects is maybe not that straightforward.
- I suggest an additional Figure marking the different flowstripes and their origin (e.g. Fig. 9 in Ely et al., JGR, 2017). If 4 flowstripes can be uniquely attributed to the basal landforms 1-4, then this would strengthen the paper. Do the flowstripes have a characteristic end-point somewhere farther upstream, potentially marking the along-flow extent of the basal landforms?

Radar reflectivity analysis: I appreciate the interpretation of the radar amplitudes for the detection of potentially localized basal water. However, the point that needs to be made here is that areas of high basal reflectivity correspond with the predicted water flow paths. Therefore, I suggest to directly mark the three subglacial water channels (and their width, and preferably uncertainty) directly in the corresponding radargrams. The hydrostatic potential is in my eyes in this regard not very useful.

Comparison with previous findings: In l. 242 briefly compares the new findings to our previously published findings (Drews et al., 2016, Nat. Comm.) and suggests that 'our' features could also be interpreted as hard-bedded features. I am not against this interpretation and, in hindsight, I surely should have considered this hypothesis in more detail. What I am missing is a more in-depth comparison of the different datasets and their interpretations. I am currently unsure if we (i) see the same feature and have different interpretations, (ii) see different features, or (iii) if we focus on different aspects of the same features (i.e. hard-bed landforms which are covered by sediments). Here are some thoughts from my side about this issue, and it may be informative to explore some of these aspects:

1. Figure 1 in our paper shows that Reflectors A-C have much internal structure (I don't think those are off-angle reflectors). This is inconsistent to me with a hard-bed landform. (Maybe we are observing different features?)
2. We see some evidence, that the bumps in the surface topography linked to the basal landforms are (sometimes) advected into the adjacent ice shelves (Fig. 7). We have interpreted this as sign for "activity" of the basal landform which is inconsistent with a hard-bed feature. (Maybe we are observing different features?)

3. Our landforms are max. 300 m wide, which is 3-13 times narrower than what is observed in this paper. (Maybe we are observing different features?)
4. Our landforms are not linked by flowstripes, and are not in a setting of glacier confluence. (Maybe we are observing different features?)
5. We think (but have no hard evidence) that “our” landforms only reach their large height in a narrow (maybe 1-2 km) region very close to the grounding line. This is inconsistent with your observations which show nicely that the landforms extend much farther upstream. (Maybe we are observing different features? Maybe our interpretation is wrong?)

Small Remarks

l. 32: “that”  “than” (?)

l. 123 “They are divided in distinct sets relative to the origin of the ice”. I think the following lines only describe the ridges on the Academy Glacier. Not sure where the other “distinct set” is taken up again.

l. 233 I am lost what confirms the “zig-zag” of the grounding line, and how this relates to the large tidal uplift of 18.5 km. Does the “zig-zag” have clear correspondence in the surface topography?

l. 434 “or” – “of”

l. 434 is it not “four” (instead of “three”) channels and landforms ?

I like the terminology of using M and U channels and will use it in the future. I also very much like Supplementary Figure S4 adding good evidence for the hard-bed landform interpretation.

Figure 1: Academy Glacier, Support Force Glacier and Foundation Ice Stream are not labelled in the Figures.

Figure 1 does not convey the merging of M-channels. It is simply not visible, especially in print.

Figure 1d should additionally mark the locations of the landforms+the calculated water flowpaths, otherwise, how should the reader make the connection that these landforms guide the water flow? I also suggest to zoom much more into the survey area, I don’t think it is needed to always show the entire catchment.

Figure 2 Profiles M-M’ and N-N’ mark the trunk area referred to in l. 111.

Figure 2 Mark flow direction of ice (into our out-of-plane). Mark surface and multiples.

Figure 2 Indicate clearly if I-I’ is grounded or not.

Figure 2: How do you define the height of “mega-ridge” 2 in profile K-K’? There is different heights depending if you choose left or right side no?

Figure 2: It is probably better to plot relative to sea level not WGS84. **Double check the 0 in L-L' ? This makes no sense to me.**

Figure 2: It would be helpful to have a subplot in each transect showing the surface topography in a blown-up version. It would be nice to know if the basal bedforms are accompanied by surface bumps (marking the flowstrieps?), and where the M-channels are in relation to the U channels. If such topography is not available state that explicitly as a limitation.

Figure 2: Profile L – L' left side, I don't see four U Channels. Only three. (What is arrow 4 pointing to?)

Figure 3: Width and Height y-axis should start at 0. Mark position (or better range) of grounding-line estimates in these plots as well. (i. e. k-j grounded, h floating).

Figure S1: Don't you use an updated version of Bedmap 2? Caption does not state this. Not sure what this Figure adds to the entire story. I would prefer a zoom to the FIS area (and some measure of uncertainty as mentioned above).

Figure S2

There are no blue, orange or green lines indicating the different areas of the grounding zone. There is only one yellow line.

Reviewer #2 (Remarks to the Author):

Review Jeofry et al., Nature Comms

Jeofry et al. use imagery and radar profiles to infer large longitudinal ridges in the bedrock beneath Foundation Ice Stream, and identify how longitudinal corrugations in the ice as it goes afloat guide sub-glacially exiting water into the incised groove and create a sub-ice-shelf channel. This last process was also discussed more generally (looking Antarctic-wide) by LeBrocq et al. and K. Alley et al.

The study is clear and well written, and straightforward. It would seem to be a kind of confirmation and detailed assessment of an example of processes that have been discussed before. The scale of the bedrock ridges is interesting, I'm not aware of ridges like this in sub-aerally exposed formerly glaciated terrain. I think it could be published after some significant revisions --- in particular, improvements in the figures.

The study spends a bit of time pointing out a suture between 'East Antarctic ice' ice and 'West Antarctic ice', but there would seem to be few differences.

For the authors: What is the degree of grounding (the difference between the observed surface height and the height at which the ice would go afloat, given the thickness) for the grounding zone region? A 5m tidal amplitude is significant, but it would appear that the slope rises significantly more than that upstream of the MOA and ASAID grounding lines. I think a closer look at Brunt et al. will show that the regions where there is a significant bench, where ice is only lifted off the bed at king tides, are very nearly at the same elevation as the hinge zone and MOA or ASAID picks.

L22 ...flow and are thought...

L25 ...beneath meandering channels that couple... (no comma)

L25-26 ... As the ice transitions to floatation...

L27 I think you need to infer that there is water from the radar reflections..

L29 ...corrugations, accentuating them as flow progresses downstream?

L51 I think you could find a more pertinent reference here – Kulesa et al. is a good study, but it the process they describe is a subset of things that destabilize ice shelves and therefore (potentially) ice sheets. How about Fürst et al., 2016, or DuPont 2005, or Rignot 2004.

L136 need a comma: ... ice-penetrating radar, the...

L199 Re-write – 'are so big' and 'have an influence' – no matter what size, the wyou dhave some influence.

L202-203 -- ..separate attenuation correction for radar traces in West Antarctic and East Antarctic ice.

L215 – what is 'electrically deep basal water'? I have not heard this expression before. I'm guessing that it means a water layer thick enough to appear infinitely deep to radar?

Figures need work

Figure 1a – add locator map of Antarctica in the upper right, and add a diagram of what is meant by M- and U-channels in the lower left – then indicate where you say these M and U channels are occurring on this first image map. Add latitude-longitude graticule to Figure 1a. Fade the color saturation on velocity of Figure 1c (and use MOA as a backdrop). Indicate the location of the area of Figure 1d. Label all the major glacier tributaries and other geographic features

Figure 2 – your figure will have more impact if you use fewer cross-sections. One or two out of A-A', B-B', C-C' is enough. Remove 2 out of the series from D to H. N-N' is enough, you don't need M-M'. Do you not have laser altimetry to go with each of these profiles? This would be a major component to add, as a panel just above each profile.

Figure 3. These graphs do not hit the home run you want at the end. How about plotting the incision depth, by subtracting the thickness at the top of the U-channel from the mean thickness

on either side of the channel? I don't see the point of also plotting the surface height. You might be able to contrive a graphic that allowed you to show both width and incision height on the same graphic, with different colors. Indicate the grounding line point in the graphs.

Supplemental Information –

Figure S1 – nice map, revise last sentence:

'The maximum possible size of the sub-glacial drainage systems are shown. This assumes that the bed is at the pressure melting point and thawed at all points, which may not be true.'

Figure S2 – the several grounding lines are not shown (not in my copy of the .doc Supplement file), only the DinSAR line.

Note that a figure between S2 and S3 would be a good place to show the additional cross-section profiles that I suggested be dropped from Figure 2 in the main text.

Reviewer #3 (Remarks to the Author):

This is a potentially very useful paper with a great data set and is eminently publishable after major revision. Its weakness is that it appears to have been written in a glaciological bubble so to speak divorced from glacial geological and geomorphological work on ancient ice stream beds elsewhere that the authors should connect to. There is also remarkably little descriptive data presented in regard to a new type of fluvial channel the authors claim to have discovered on the surface (and below) the ice stream (M and U channels). The text is not helpful in creating a strong sense of what these are and a simple cartoon figure of the ice stream and its bed (and its relief and geology) is needed.

The paper is over illustrated for Nature Communications and some figures are clearly of less importance and can be deleted. There is some awkward written English in places and statements that do not make sense from a glacial geological or geomorphological perspective.

The strongest parts of the paper are the excellent radar images of the hard bed below the ice stream and their value should be maximised by making reference to recent publications in Pleistocene glacial geology where the topic of ancient 'hard bedded ice streams' is a major focus of activity. Ice streams were once thought to be limited to soft beds but are increasingly being recognized on hard beds such as crystalline Shields in North America and Scandinavia and flanking Paleozoic carbonates. The advantage of this work is that the former hard beds of ice streams are well exposed and can be mapped in detail and their relationship with 'mixed' and 'soft' beds directly observed. The statement in the Abstract that 'hard-bedded glacial geomorphological landforms influence subglacial hydrology and ice-shelf structure, explaining the origin of meandering surface channels, and their influence on ice-sheet dynamics could be more widespread than thought previously' is rather obvious and naive in the light of a large body of Pleistocene work elsewhere and cannot in any way be portrayed as a new observation. I think this is simply that the authors are glaciologists, not glacial geologists or geomorphologists. Their lack of geological expertise is revealed by this curious statement in the Abstract that hard beds are 'passive recorders of change.' They are in fact rather dynamic surfaces and undergo substantial erosion and sculpting by direct glacial abrasion and meltwaters and their contained abrasive sediments. The elongated bedforms described here and typical of hard beds elsewhere are drumlinized and megalinedated surfaces no different from that found on soft beds and thus are anything but 'passive.' The authors may wish to consult the following for information on hard beds below ice

streams.

These may be of use:

Eyles, N. and Doughty, M. 2016. Glacially-streamlined hard and soft beds of the paleo-Ontario Ice Stream in central Canada. *Sedimentary Geology* 338, 51-71.

Eyles, N. and Putkinen, N. 2014. Glacially-megalinedated limestone terrain of Anticosti Island, Gulf of St. Lawrence, Canada: onset zone of the Laurentian Channel ice stream. *Quaternary Science Reviews* 88, 125-134.

Eyles, N. 2012. Rock drumlins and megaflutes of the Niagara Escarpment, Ontario, Canada a hard bed landform assemblage cut below the Saginaw-Huron Ice Stream. *Quaternary Science Reviews* 55, 34-49.

Livingstone, S.J., O'Cofaigh, C.O. and Stokes, C.R., Hillenbrand, C-D., Vieli, A., Jamieson, S.S.R., 2012. Antarctic paleo-ice streams. *Earth-science Reviews* 111, 90-128.

Ó Cofaigh, C., Pudsey, C.J., Dowdeswell, J.A. and Morris, P. 2002. Evolution of subglacial bedforms along a paleo-ice stream: Antarctic continental shelf. *Geophysical Research Letters* 29, 1199, 10.1029/2001GL014488, 2002.

Cross-referencing to other work would be most valuable in the paragraph spanning pp. 3-4 where bedrock ridges are described. This needs some information of the likely geology of the bed which as it is presented here, is only very briefly alluded to be reference to the geology of the bed of nearby Dotson Ice Shelf. The use of the term 'hard-bed flow-parallel landforms' throughout the paper is ambiguous: are they rock drumlins or roche moutonnées? What is their length parallel to flow?

The text descriptions of the so-called M and U channels are not useful and the reader does not come away with a clear idea of exactly what they are nor the processes that formed them or their glaciological significance. These require full description and a short publication in *Nature Communications* is perhaps not the place to do this. A simple cartoon figure showing the surface of the ice stream and what the authors infer to be going on at its surface and at its underlying bed, would be very helpful.

It is not easy to see M channels on Figure 1 and they are illustrated on no other figure. Are they truly meandering or sinuous? The correct terminology is very important in fluvial geomorphology. There is remarkably little information presented in the paper on these channels yet the authors wish to introduce a new term ('M channel') to the literature. They need to be much better defined and illustrated. Again, a simple three dimensional cartoon figure would help.

Figure 1 is a multi-part figure (a-d) and by the time this is published it will be too small to see important details. Figure 1 b refers to 'basal elevations' which presumably refers to 'bed topography' but it does not show the topography of much of the ice stream in question, and in particular the area of M channels under investigation. This is a curious omission.

In general, the paper is over-illustrated and a revised version could reduce the number of radar transects by 60%. They all show the same thing so chose representative panels; *Nature Communications* is a 'short format' journal. What are the panel diagrams in Figure 3 titled 'along track distance' supposed to be showing? I suggest delete these; the radar profiles are the visual strength of the paper so emphasize those.

The same concerns surround recognition of subglacial 'U channels.' Are these true erosional channels or simply narrow topographic lows floored by softer strata between more resistant

bedrock highs? What is the geology of the bed? Is it strongly foliated (in the case of crystalline strata) or otherwise stratified (sedimentary) that would allow glacial abrasion to produce an irregular corrugated surface? Again, all important geological information is missing.

To summarize this is a potentially useful paper; the radar images of the ice stream bed are very informative but they do not alone, warrant publication. Much better basic information is needed on the form of the so-called M and U channels before their origin(s) is entirely believable, and cross-referencing to other work elsewhere would be very helpful. A simple diagrammatic cartoon of the ice stream and shelf, its bed and generalized geology would help as would deletion of all non-essential or duplicative figures.

It has been a pleasure reading this paper.

N. Eyles

Toronto

Referee #2 (anon.)

Jeofry et al. use imagery and radar profiles to infer large longitudinal ridges in the bedrock beneath Foundation Ice Stream, and identify how longitudinal corrugations in the ice as it goes afloat guide sub-glacially exiting water into the incised groove and create a sub-ice-shelf channel. This last process was also discussed more generally (looking Antarctic-wide) by LeBrocq et al. and K. Alley et al. The study is clear and well written, and straightforward. It would seem to be a kind of confirmation and detailed assessment of an example of processes that have been discussed before. The scale of the bedrock ridges is interesting, I'm not aware of ridges like this in sub-aerially exposed formerly glaciated terrain. I think it could be published after some significant revisions --- in particular, improvements in the figures.

We thank the referee for this positive evaluation. We have worked hard to improve some of the text and in particular the figures, as discussed below.

The study spends a bit of time pointing out a suture between 'East Antarctic ice' ice and 'West Antarctic ice', but there would seem to be few differences.

The glaciological 'suture' between east and west Antarctica is an important finding of our work – not the fact that it happens, as that is obvious from surface velocities and surface flowstripes, but how this manifests englacially and subglacially, which has not been shown previously. We find the bed conditions are quite different, revealing the Foundation ice stream to be very much a significant outlet with adjoined yet discrete glaciological systems. This is not a major point of the paper, but it is glaciologically fascinating to understand.

For the authors: What is the degree of grounding (the difference between the observed surface height and the height at which the ice would go afloat, given the thickness) for the grounding zone region? A 5m tidal amplitude is significant, but it would appear that the slope rises significantly more than that upstream of the MOA and ASAIID grounding lines. I think a closer look at Brunt et al. will show that the regions where there is a significant bench, where ice is only lifted off the bed at king tides, are very nearly at the same elevation as the hinge zone and MOA or ASAIID picks.

This is a good point to make. With the data we have, we are able to determine the sensitivity of the grounding line to tides using the along-flow radar transect NN'. We use the standard flotation equation, as used for example in Ross et al. (2012, Nature Geosciences). We see that a 5 m rise in sea level can shift the grounding line upstream by over ~10 km. Hence, through further investigation we are able to support our original proposition that the ice-sheet response to tides may well be able to explain the various grounding lines of Foundation ice stream.

This finding also means that a large section (between the grounding lines demarked from flexure vs. slopes) of the Foundation ice stream trunk is lifted off its bed with the tides. We have added this information to Figure 1 NN', and discussed it in the text. We note that this analysis is possible only because of the radar data available in section NN'.

We have also undertaken a new analysis of the basal reflections along the trunk of the ice stream (primarily for Referee #1, see later), in which we confirm the basal properties downstream of the Rignot grounding line transition to that of floating ice. This transition occurs around the location of our flotation calculation, and between those determined by flexure and slopes analysis. Again, this information is now included in Figure 1 NN'.

Based on these new analyses, we conclude there to be an extended 'grounding zone' between the flexure and slope driven grounding lines; a finding consistent with the idea that ice-sheet grounding lines are far more complex than thought previously. We believe this finding is significant, and will be of value to the wider glaciological community. We thank the referee (and Referee #1) for suggesting we look more deeply into the data available to us.

This new work has allowed us to re-inspect the very large (800m) basal feature we had previously interpreted as being a U-channel. This feature is located within the 'estuary' part of the grounding zone. To fully understand whether this feature was a U-channel, or in fact a rock pinnacle, we analysed the reflectivity off its crest relative to the flat region of the ice-stream base (between landforms), which has reflectivities consistent with water. Our calculations reveal the crest of the basal feature to be 8.1 dB higher in reflectivity than the wet ice-stream bed. This is consistent with deep (>10 m) water, as would normally be observed over a large subglacial lake. Hence, we are confident that the feature is a water-filled U-channel (at least at its crest). The explanation for why the ice surface is not depressed above it is due to 'bridging stresses' from the neighbouring semi-grounded ice, and also because the feature is a slight side-echo from downstream, meaning the aircraft altimetry above it will be misleading.

Importantly, a corresponding (same flowline) bedform adjacent in Line L-L' shows the feature to be only 400 m in height. Thus, between the two transects the bed feature increases in height substantially down flow. This is consistent with the theory of well-organised subglacial water development of U-channels (though has never been observed previously). We now mention this explicitly in the paper.

We thank the referee for encouraging us to look more closely at this feature – the resulting insights have improved the paper substantially.

Minor edits

L22 ...flow and are thought... *Changed.*

L25 ...beneath meandering channels that couple... (no comma) *Changed.*

L25-26 ... As the ice transitions to floatation... *Changed.*

L27 I think you need to infer that there is water from the radar reflections.. *This has been done (see paragraph above), and we have modified the abstract accordingly*

L29 ...corrugations, accentuating them as flow progresses downstream? *Modified as requested*

L51 I think you could find a more pertinent reference here – Kulesa et al. is a good study, but it the process they describe is a subset of things that destabilize ice shelves and therefore (potentially) ice sheets. How about Fürst et al., 2016, or DuPont 2005, or Rignot 2004. *Many thanks. We have included Fürst et al., 2016.*

L136 need a comma: ... ice-penetrating radar, the... *Added.*

L199 Re-write – 'are so big' and 'have an influence' – no matter what size, the wyou dhave some influence. *Agreed – and modified as necessary. We have rewritten this section, to focus more on the radar reflectivity first.*

L202-203 -- ..separate attenuation correction for radar traces in West Antarctic and East Antarctic ice. *changed.*

L215 – what is 'electrically deep basal water'? I have not heard this expression before. I'm guessing that it means a water layer thick enough to appear infinitely deep to radar? *We have changed this description to read ~5 m deep.*

Figures need work

Figure 1a – add locator map of Antarctica in the upper right, and add a diagram of what is meant by

M- and U-channels in the lower left – then indicate where you say these M and U channels are occurring on this first image map. Add latitude-longitude graticule to Figure 1a. Fade the color saturation on velocity of Figure 1c (and use MOA as a backdrop). Indicate the location of the area of Figure 1d. Label all the major glacier tributaries and other geographic features.

This has been changed, as requested.

Figure 2 – your figure will have more impact if you use fewer cross-sections. One or two out of A-A', B-B', C-C' is enough. Remove 2 out of the series from D to H. N-N' is enough, you don't need M-M'.

We understand that this figure may involve a lot of transects, but since Nature Communications offers space for expanded figures, and since they are important to the paper, we'd prefer to keep them all together at this stage (we're happy to take an editorial decision, however). Do you not have laser altimetry to go with each of these profiles? This would be a major component to add, as a panel just above each profile. *We have added aircraft altimetry over the transects, as requested.*

Figure 3. These graphs do not hit the home run you want at the end. How about plotting the incision depth, by subtracting the thickness at the top of the U-channel from the mean thickness on either side of the channel? I don't see the point of also plotting the surface height. You might be able to contrive a graphic that allowed you to show both width and incision height on the same graphic, with different colors. Indicate the grounding line point in the graphs.

We have moved the graphs to Supplementary Information, and have moved the map of glacial geology into the main paper (in line with Referee #1 comments).

Supplemental Information –

Figure S1 – nice map (*thanks*), revise last sentence:

'The maximum possible size of the sub-glacial drainage systems are shown. This assumes that the bed is at the pressure melting point and thawed at all points, which may not be true.' *Changed.*

Figure S2 – the several grounding lines are not shown (not in my copy of the .doc Supplement file), only the DinSAR line. *This change has been made.*

Note that a figure between S2 and S3 would be a good place to show the additional cross-section profiles that I suggested be dropped from Figure 2 in the main text.

As mentioned above, we'd prefer to keep all the radargrams together at this stage.

Referee #3 (Eyles)

This is a potentially very useful paper with a great data set and is eminently publishable after major revision. Its weakness is that it appears to have been written in a glaciological bubble so to speak divorced from glacial geological and geomorphological work on ancient ice stream beds elsewhere that the authors should connect to. There is also remarkably little descriptive data presented in regard to a new type of fluvial channel the authors claim to have discovered on the surface (and below) the ice stream (M and U channels). The text is not helpful in creating a strong sense of what these are and a simple cartoon figure of the ice stream and its bed (and its relief and geology) is needed. The paper is over illustrated for Nature Communications and some figures are clearly of less importance and can be deleted. There is some awkward written English in places and statements that do not make sense from a glacial geological or geomorphological perspective. *We accept that the paper can be edited in both text and figures, to tighten our arguments. We thank the referee for also acknowledging the potential strengths of the paper, however.*

The strongest parts of the paper are the excellent radar images of the hard bed below the ice stream and their value should be maximised by making reference to recent publications in Pleistocene glacial geology where the topic of ancient 'hard bedded ice streams' is a major focus of activity. *This is an excellent point, and we have aimed to do this in the revision.*

Ice streams were once thought to be limited to soft beds but are increasingly being recognized on hard beds such as crystalline Shields in North America and Scandinavia and flanking Paleozoic carbonates. *We agree!* The advantage of this work is that the former hard beds of ice streams are well exposed and can be mapped in detail and their relationship with 'mixed' and 'soft' beds directly observed. The statement in the Abstract that 'hard-bedded glacial geomorphological landforms influence subglacial hydrology and ice-shelf structure, explaining the origin of meandering surface channels, and their influence on ice-sheet dynamics could be more widespread than thought previously' is rather obvious and naïve in the light of a large body of Pleistocene work elsewhere and cannot in any way be portrayed as a new observation. *It can be in terms of extant ice streams, however, and this is the value of the paper – linking palaeo observations to modern ice sheet processes.* I think this is simply that the authors are glaciologists, not glacial geologists or geomorphologists. Their lack of geological expertise is revealed by this curious statement in the Abstract that hard beds are 'passive recorders of change.' They are in fact rather dynamic surfaces and undergo substantial erosion and sculpting by direct glacial abrasion and meltwaters and their contained abrasive sediments. The elongated bedforms described here and typical of hard beds elsewhere are drumlinized and megalineated surfaces no different from that found on soft beds and thus are anything but 'passive.' The authors may wish to consult the following for information on hard beds below ice streams. *To clarify, what we mean is that the landforms are active in the development of the ice stream down-flow – we realise that they must be dynamic evolving features of course, but over different timescales than hydrological and glaciological processes. We find it important to recognise that processes that have acted on the landscape development many thousands of years ago have an impact today on ice-shelf processes.*

We thank the referee for these insights, and confirm that we have now made changes to the abstract and text, as recommended. We feel it is greatly improved as a consequence.

These may be of use:

Eyles, N. and Doughty, M. 2016. Glacially-streamlined hard and soft beds of the paleo-Ontario Ice Stream in central Canada. *Sedimentary Geology* 338, 51-71.

Eyles, N. and Putkinen, N. 2014. Glacially-megalineated limestone terrain of Anticosti Island, Gulf of

St. Lawrence, Canada: onset zone of the Laurentian Channel ice stream. *Quaternary Science Reviews* 88, 125-134.

Eyles, N. 2012. Rock drumlins and megaflutes of the Niagara Escarpment, Ontario, Canada a hard bed landform assemblage cut below the Saginaw-Huron Ice Stream. *Quaternary Science Reviews* 55, 34-49.

Livingstone, S.J., O’Cofaigh, C.O. and Stokes, C.R., Hillenbrand, C-D., Vieli, A., Jamieson, S.S.R., 2012. Antarctic paleo-ice streams. *Earth-science Reviews* 111, 90-128.

Ó Cofaigh, C., Pudsey, C.J., Dowdeswell, J.A. and Morris, P. 2002. Evolution of subglacial bedforms along a paleo-ice stream: Antarctic continental shelf. *Geophysical Research Letters* 29, 1199, 10.1029/2001GL014488, 2002.

We thank the referee for these suggestions, and have included them in the paper. We have also included former Supplementary Figure S4, and the references cited in its caption, into the main paper (new Figure 3), to make the link between our work and previous work on hard-bedded glacial landforms more obvious.

Cross-referencing to other work would be most valuable in the paragraph spanning pp. 3-4 where bedrock ridges are described. This needs some information of the likely geology of the bed which as it is presented here, is only very briefly alluded to be reference to the geology of the bed of nearby Dotson Ice Shelf. The use of the term ‘hard-bed flow-parallel landforms’ throughout the paper is ambiguous: are they rock drumlins or roche moutonnées? What is their length parallel to flow?

We do not have sufficient spatial resolution in the data to determine whether the landforms are rock drumlins or roche moutonnées. We are confident that they are flow-parallel landforms, due to bed roughness perpendicular to flow and bed smoothness along ice flow, and while we can constrain their widths in cross-flow data, we are unable to determine their alongflow profiles. We are also confident that the features are hard bedded though comparison of their width/heights compared to known hard bedded flow-parallel features measured over formerly glaciated terrain in Antarctica.

The text descriptions of the so-called M and U channels are not useful and the reader does not come away with a clear idea of exactly what they are nor the processes that formed them or their glaciological significance. These require full description and a short publication in *Nature Communications* is perhaps not the place to do this. A simple cartoon figure showing the surface of the ice stream and what the authors infer to be going on at its surface and at its underlying bed, would be very helpful.

We have included a simple cartoon within Supplementary Information, as suggested.

It is not easy to see M channels on Figure 1 and they are illustrated on no other figure. Are they truly meandering or sinuous? The correct terminology is very important in fluvial geomorphology. There is remarkably little information presented in the paper on these channels yet the authors wish to introduce a new term (‘M channel’) to the literature. They need to be much better defined and illustrated. Again, a simple three dimensional cartoon figure would help.

We agree – M channels could be better presented. We have contrast-stretched the MODIS images to make them much clearer.

Figure 1 is a multi-part figure (a-d) and by the time this is published it will be too small to see important details. Figure 1 b refers to ‘basal elevations’ which presumably refers to ‘bed topography’ but it does not show the topography of much of the ice stream in question, and in particular the area of M channels under investigation. This is a curious omission.

The bed topography is best examined in the radar profiles – all of the digital elevation maps of subglacial Antarctica (and Greenland) suffer from interpolation issues that can blend out the details of landforms. This is a problem across the whole of Antarctica. While such DEMs are of great value

for broad views of the bed, and water flow, the details must still come from discrete radar transects (or, rarely, high-resolution radar grids).

The best digital elevation model of the entire region is in Jeofry et al. (ESSD, 2018), which we use in this paper and refer to.

In general, the paper is over-illustrated and a revised version could reduce the number of radar transects by 60%. They all show the same thing so chose representative panels; Nature Communications is a 'short format' journal. What are the panel diagrams in Figure 3 titled 'along track distance' supposed to be showing? I suggest delete these; the radar profiles are the visual strength of the paper so emphasize those. *We have moved the analysis of bedform size to the SI, but we would prefer to keep all of the radargrams together. While Nature Communications is a short format, it also offers opportunity for expanded figures by being online. Breaking up the radargrams might confuse the readers, by making them go back and forth between the paper and SI.*

The same concerns surround recognition of subglacial 'U channels.' Are these true erosional channels or simply narrow topographic lows floored by softer strata between more resistant bedrock highs? *Yes, the latter is our opinion, and we now make it clear in the paper.* What is the geology of the bed? Is it strongly foliated (in the case of crystalline strata) or otherwise stratified (sedimentary) that would allow glacial abrasion to produce an irregular corrugated surface? Again, all important geological information is missing. *I'm afraid we cannot know the answer to this. We don't believe not having it seriously affects the main results of the paper, however.*

To summarize this is a potentially useful paper; the radar images of the ice stream bed are very informative but they do not alone, warrant publication. Much better basic information is needed on the form of the so-called M and U channels before their origin(s) is entirely believable, and cross-referencing to other work elsewhere would be very helpful. A simple diagrammatic cartoon of the ice stream and shelf, its bed and generalized geology would help as would deletion of all non-essential or duplicative figures. It has been a pleasure reading this paper. *We thank the referee for these positive concluding remarks.*

Referee #1 (Drews)

Jeofry et al. present new observations from radar data covering the Foundation Ice Stream in West Antarctica. They use modelling of the subglacial hydrology in conjunction with interpretations of the basal reflectivity to draw their conclusions. The latter can be summarized as follows:

1. Ice-shelf channels (curvilinear tracks of thin ice which frequently occur on Antarctic ice shelves) in the ice shelf connected to the FIS are induced by topographic undulations located upstream of the grounding line. *The ice sheet is corrugated by the hard-bed landforms, but the ice-shelf channels are formed by subglacial water exiting the ice sheet and feeding upwards into the top of the corrugations, once the ice is afloat. Hence, the referee's summary of the finding isn't quite right. The ice shelf channels are formed by basal water that preferentially feeds upwards into the corrugation caused by ice flow along hard-bedded flow-parallel glacial geological features.*
2. The topographic undulations are flow-parallel, hard-bed landforms (2–3 km wide; 100-200 m high; 10s of kilometers long; spacing 1 km; potentially covered by till). Implicitly, it is concluded that the hard-bed landforms cause formation of flowstripes at the surface. *We do not implicitly conclude that the landforms cause the flowstripes. We assume (reasonably) that both landforms and flowstripes are aligned parallel to ice flow, but make no comment on their co-genesis.*
3. The flow-parallel, hard-bed landforms determine the location of waterfilled, subglacial channels which terminate (at the grounding line) exactly where ice shelf channels are observed. The U channels are thus further incised upwards through localized, plume-driven melting. *Yes, we believe this is reasonable to conclude from our data. We can't see an alternative explanation. We don't see this as contentious.*
4. The joint observations also explain the meandering/merging of ice-shelf channels farther downstream. *Yes, our data explicitly confirms this merging takes place in the surface imagery and at the ice shelf base from the RES data. This should not be controversial.*
5. The grounding line of FIS is lifted up on tidal cycles explaining the differences of 18.5 km in published grounding-line datasets. *We originally offered this as an explanation, but not a firm conclusion. With new analysis of the flotation (see earlier comments from Referee #2), we are now able to state this with confidence. Hence, our further analysis has allowed us to bolster this conclusion and we thank the referee for pushing us to inspect the data more closely.*

This paper tackles an important point in glaciology linking to the stability of Antarctic ice shelves and their role in buttressing the upstream ice flux. There is currently no consensus if ice-shelf channels stabilize or destabilize Antarctic ice shelves, and this is partially due to their unknown origin. This paper has potential to investigate this matter further. *We thank the referee for these supportive remarks. We now add a sentence to make the connection between our work and potential future ice shelf stability – noting that if future surface melting took place, it would be routed within M-channels which could incise them further.*

General Remarks

The findings confirm some aspects (i.e. conclusions 1 & 3) in a study that we published earlier (Drews et al., Nat. Com. 2017) but also provide alternative explanations (i.e. conclusions 2 & 4). The latter may apply to our previous study (as alluded in l. 243), but could also describe a different phenomenon. Although I provide much criticism in the comments below, I want to stress that I am not defensive against the alternative explanations. All comments are fully geared to make this paper more solid and also more easy to follow. Unfortunately, I find the paper currently imprecise in many places. Arguments are not consistently applied, and not enough information is at hand to check their validity. I have some confidence in conclusions 1 and 2, but am unconvinced by conclusion 3-5.

Therefore, I don't think that the paper can be published in its current form. Below I have some questions/remarks which I hope the authors will find useful for improving the current version. *We thank the referee and completely accept that in responding to them we can make the paper better.*

Specific Remarks from Referee #1

Subglacial water routing and its relation to the basal landforms:

The subglacial hydrology routes based on the updated thickness model show now three instead of one water flowpath previously. As I understand it right now, the authors only updated the ice thickness for their novel calculations, but not the surface elevation of Bedmap2 (which is outdated by now). The following questions come to mind:

1. Wright et al. (Geophys. Res. Lett., 2008) show that the surface topography is very important for calculating the subglacial drainage network. Is this not the case here and why not? *Yes, this is the case, but where the ice surface is very flat (at and upstream of the grounding line) the bed will have a more dominant influence than it would upstream where ice surface gradients are steeper. We have used Cryosat2 ice elevations – the newest product – with our new bed elevation to get the best possible idea of basal water flow, and can confirm the water is expected to be routed along the bedform direction. However, we note this is just a model; the best idea about the presence of water comes from the radar reflectivity measurements. They are consistent and provide a compelling indication of water is routed beneath the ice, exits the ice sheet and affects the ice shelf.*

That said, we have now included Wright et al. 2008 in the paper in our discussion around the issue.

2. What resolution does the surface DEM have? *The DEM is taken from Jeofry et al. (ESSD, 2018). The resolution of the DEM is ~1 km. The mass conservation interpolation of the gridded data adds further refinement, improving accuracy if not resolution.*

However, as we say above, the water routing is just output from a model. Water flow should be examined in view of the radar reflectivity also (indeed this is direct evidence, whereas a model is a simplification of processes). From analysis of RES reflectivity in cross-flow radargrams, we are confident that the water flows alongside the landforms and in the line of ice flow.

3. Do the flowstripes have a correspondence in the surface topography, and if so, would that impact the prediction of the subglacial drainage routes? *No, the surface flowstripes have negligible influence on the surface profiles. The M-channels do, however, and we have now included the surface altimetry over them alongside the radargrams.*

4. How different are the predicted flowpaths when using a gridded vs. a mass-conserving bed? This is important, because I suspect that the gridded version does not make an along-flow connection between the subglacial landforms observed in the radar transects J-J', K-K'. However, I presume that such a connection is required if the basal topography should guide the subglacial water flow. I don't know what gridding + mass conservation changes in that respect. Outsourcing these points to the "Jeofry et al., in press" reference only works if those points are discussed in detail there. *This is a good point, and there are some differences to the upstream drainage. But, in all cases, water is routed along the bed of Foundation ice stream trunk in line with the ice flow direction, which is the same as both flowstripes and bedforms. We choose the 'mass conservation bed' because it does not have the artefacts inherent in Bedmap2 that can affect flow routing, especially where the surface elevation is flat.*

However, we reiterate that the most conclusive evidence of the precise location of water comes from the cross-flow radargram reflectivities, and in our revised paper we now interrogate these data more closely.

I understand that those are quite detailed questions, but at the same time, the authors draw detailed conclusion from the analysis (i.e. the predicted flowpaths are reliable on 1 km scales, which is the scales of the landforms/M/U channels). Without a sensitivity analysis of the subglacial water routing as a function of the input variables, I don't see how this point can be justified. (I apologize if I missed essentials of the subglacial hydrological modelling, of which I am not a specialist). *This is a good point, and we are pleased to add further substantiation along the lines above. However, the main point of the paper, that the bed landforms are too big to be sedimentary (by analogy with known landforms and lack of evidence in the geological record for similar sized sedimentary landforms) and that water is routed alongside them (as opposed to across them), is fully in line with the data presented and should be uncontentious.*

Moreover, I currently don't see improvement to the results which have already been published by LeBrocq et al., Nat. Geosc., 2013. The suggested improvement is that the subglacial landforms will "determine the position of any (sic!) buoyant subglacial water emerging from the grounding zone.." (l. 234). Yet, "Only the smaller 'East Antarctic' basal water route is aligned with flow-parallel bedforms." (l. 175). Those are directly conflicting statements. *We are not improving the paper by simply identifying more ice-shelf channels. As the referee says, this has been done before. What has been missing is an assessment that hard-bedded morphology is implicated in their formation and location at Foundation. Hard bedded basal morphology is well known from previous work in formerly glaciated terrain (as Referee #1 points out), but has been surprisingly lacking among contemporary glaciology. Here, we make an important link between these communities. This is a new finding, and we believe because of the likely presence of hard-bedded morphology elsewhere will be similarly influential in other parts of Antarctica (and former ice sheets).*

We disagree that our statements are in conflict however. The two units of the ice stream have different bed characteristics, as our radar reveals. We do not see this necessarily as being surprising. But in any case, are data simply reflect that there are differences in bed roughness and basal water conditions. Only across the East Antarctic side do the very large bedforms exist. We don't believe this is conflicting, when one views the data. It is unusual, however, and we feel the community should recognise that ice streams sourced from different upstream catchments can have distinct and separate bed conditions as a consequence of their origins.

*Our paper adds to the growing understanding of **complexity** beneath ice streams and their grounding lines.*

On the hydrology, we must confess that our initial analysis of water routing was incorrect. We have now made the necessary changes, and confirm that the bulk of East Antarctic water is routed through the trunk of the ice stream, and also that West and East Antarctic subglacial water remain separated. This makes a lot more intuitive sense, and we thank the referee for bringing this issue up.

(Also, if the "East Antarctic" route is aligned with multiple landforms (plural), does it mean it crosses the landforms 1-4?). *No. Our data do not reveal any basal water crossing over bedforms. Because of our closer analysis of the radar data, we now mention this explicitly in the paper.*

None of the Figures convey the link between the subglacial water routes and the landform sufficiently (e.g. In Figure 2b the exit points at the grounding line are overprinted by the radar profile

lines and the subglacial landforms are not marked, nor are the relevant flowstripes (there are many).). Also the M-channels are only marked at one location, but the underlying image does not allow to trace the flow stripes back to the grounding line. As a last point, the radar reflectivity analysis in the transects J-J' and K-K' should be linked to the projected water flow paths of Figure 1b, not the hydrological potential. I don't see how people are suspected to infer the most likely water flow paths from the transects of the hydrological potential and how these related to the radar reflectivity values. Given all the points I mentioned above, I hope you understand my reservations about how these findings are portrayed. *We do understand the scepticism, based on the figure material submitted, and we have worked hard to improve this in the new submission. We totally recognise that the referee is helping us make the scientific case clearer. Actions taken on these points include:*

- 1. We have made an enlarged figure of the grounding zone, where the flowstripes, M-channels and grounding lines can be seen in finer detail.*
- 2. On this image, we locate the peaks of the basal landforms, revealing their spatial alignment with surface flowstripes.*
- 3. On this image we also show the bed reflectivity, showing where basal water exists in relation to the landforms and M/U channels downstream.*
- 4. We have removed labelling of ice-stream bed features that do not influence U/M channels, to make it clearer about which bedforms are having the most impact.*

This figure is a representation of the data, with no influence from the water flow model or bed DEM.

Further incision of ice-shelf channels by buoyant melt water plume: In multiple places, the authors suggest that U channels are further incised by melt-water plumes seawards of the grounding line. There is already published evidence for this (e.g. LeBrocq et al. 2013; Marsh et al., GRL, 2015).

1. What is the additional value of this study in this regard? *We are able to explain why a subglacially-emerging source of water would preferentially upward erode an ice-shelf channel, rather than mix and spread laterally, due to ice-sheet bed corrugation at the grounding line, and buoyancy of the basal water vs ice shelf cavity water. The number of ice shelf channels we see correlated with the particularly rough bed, offering several opportunities for pre-selected upwards incision of the water. We also reveal the morphology of a particularly long channel, and reveal it to coalesce with its neighbour.*

2. If plume melting was very relevant, wouldn't you expect growing of the U-channels seawards? *No. With distance from the grounding line, we expect the plume to mix with water beneath (which may be linked with tidal flow) and reduce in temperature, hence its affect will diminish away from the grounding line, as we show. We now mention this in the paper.*

3. Instead, "U-channels remain relatively consistent across the grounding line and down flow" (l. 157). How does this fit together? *The widths remain constant, and so does the elevation. But as the ice shelf thins, the height of the channels diminish. (This is simply what our data reveal).*

From my point of view, you could argue that ice-shelf channels would close without plume melting due to lateral inflow (Drews, TC, 2015). *This is a fine paper, of course, but that would lead to the channels closing laterally – that's simply not what our data show. [We also have data from other U-channels downflow from the Moller and Institute ice streams showing the same thing].*

I apologize for advertising my own reference here, but something needs be done to strengthen the inference about plume melting and its relation to the topographically induced perturbations from farther upstream. Otherwise I don't see the additional value compared to the previous findings. *No*

need to apologise. We hope our clarifications make it clearer what we are revealing, and we accept these questions are appropriate and necessary.

The meandering of M/U-channels: The abstract states that this analysis “explains the origin of meandering surface channels” (l. 31). I don’t see progress in this paper explaining the meandering of the channels on the floating ice shelves. In fact, l. 178 explicitly states that the authors cannot distinguish whether the merging of surface channels has occurred on the ice shelf or at the grounding line. So how is the meandering then explained? *The origin of the meandering channels is explained by the paper, in that we show water flow influenced by landforms along the line of ice flow, which acts to (1) pinpoint the exit of the water; (2) corrugate the ice base once afloat; and (3) lead to the exiting water to flow up the corrugation, eroding it further downstream. The referee is right that we stated that we could not say if the merging channels are formed where they are now, as the ice shelf is constantly advecting. While we can’t rule out this might be a consequence of grounding line changes, the fact the landforms are elongated along flow means that grounding line change would not act to merge channels itself. We have now changed the paper, on reflection of this point, to make it now clearer – which also underlines the explanation of the meandering channels.*

The location of the grounding-line: The authors suggest that the grounding line is lifted up during high-tide over an along-flow section of at least 18.5 km. This supposedly explains the differences between the DInSAR grounding-lines and other grounding-lines mapping the break in surface slope (l. 117). I am not aware of any studies that have shown that tidal uplift occurs over such long distances, and this should be put into context with other studies (e.g. K. Christianson et al. in publications of the Whillans Ice Stream; or Konrad et al., Nature, 2017). However, the biggest problem for this paper is that it remains unclear if the profile I-I’ then images a floating or a grounded section of the ice sheet. If it is the former, the radar reflectivity analysis of this transect should detect basal water everywhere. However, it doesn’t, so I am left with the expression that it is grounded. Does that mean the profile was collected during low-tide? Line 154, on the other hand, describes the feature in the I-I’ transect as an 800 m U channel (and not as a flow-parallel landform), this makes me think again that you consider the U-channel to be water filled (i.e. floating?). What does this then mean for the analysis of basal reflectivity which does not detect basal water everywhere?

The position of the grounding line is an important aspect of this paper. To be clear, there are serious discrepancies between the various satellite methods to calculate the grounding line of Foundation ice stream. We are the first to inspect these discrepancies in Foundation Ice Stream with radar data, and it reveals some hitherto unappreciated facts about the various techniques and their position in the ice sheet. The radar line most pertinent to this is Line N-N’, which is along the axis of the ice stream and thus parallel to ice flow. In it, we show that the ice sheet has a classic profile across the grounding line, and that neither of the techniques for identifying it from satellite data place it in the position a glaciologist would from consideration of that profile. The ‘slope’ technique places the grounding line downstream of it. The ‘flexure’ technique places the grounding line upstream of it.

We now include a measure of the reflectivity abruptness of the ice sheet bed along N-N’ – a good determinant of grounded (low abruptness) vs floating (high abruptness) ice. Here, the abruptness changes sharply between the two sets of grounding lines, but slightly closer to the surface-slope-derived line than the flexure line. Hence, both satellite-derived grounding lines seems to be prone to some error. This work shows that care should be taken in identifying grounding line locations from satellite data, and that radar transects are needed to constrain the findings.

However, as the referee points out, we can’t necessarily say that either grounding line is wrong if tidal influences are high. Conceptually, both could be right; one at high the other at low tide. To

investigate this further, as in our response to Referee #2, we used the ice/bed elevations within N-N' to understand where the ice is afloat, and what difference +/- 5 m sea level would make to the grounding line of that profile. The analysis reveals that with a 5m tidal range the position of the grounding line can change by around 20km. Hence, we believe we have found a resolution that can explain the large difference in grounding lines for the Foundation Ice Stream. We have now included this assessment of the grounding line in the N-N' radargram, and thank the referee for stimulating further work.

The referee is correct to point out that we regard the 800 m high feature in I-I' to be water filled (from specific reflectivity analysis, Supplementary Information 2, Figure S6) above the bedform, representing the onset of the U-channel. The above analyses show that the feature initiates immediately downstream of the grounding zone (ie. at high tide the ice will likely float).

Our conclusion, based on this new analysis, is that Foundation Ice Stream contains a significant tidal-influenced 'estuary' grounding zone, and the U-channel is initiated within this zone.

I also disagree with the statement that there is no reason to doubt the DInSAR grounding line (I. 116). You can only say this, if you know which interferograms were used to detect the grounding line at this location. Some of them were not double-differenced, and even if they were, it is not always trivial to detect the landward limit (it depends on the differential tidal uplift between the image acquisition, and this can be zero in some cases). The authors must provide much more evidence to justify their statement about an >18.5 km grounding line migration by tides, only relying on previously published datasets is not enough. *Our aim is simply to point to the differences between the grounding lines and also to the fact that the location where differences occur is coincident with the channels observed. We hope the clarification given above suitably addresses these points.*

One of the most straightforward ways to do this would be to attempt to locate the grounding line in the available radar data using the basal reflectivity of the along-flow transect N-N'. This seems much easier than to detect localized melt water channels in the across-flow transect, and I wonder why this was not done. *Many thanks for this suggestion. As you can see above, it was very useful – we should have done this originally.*

Relation of flowstripes and basal landforms: The basal landforms detected in K-K' and J-Jare linked using the flowstripes. This is ok if (but only if), the flowstripes are caused by the bedforms. *We disagree. We believe both landforms and flowstripes are aligned with ice flow, but for different reasons. They don't need to have the same process of formation.*

Glasser & Gudmundsson (T. Cryosph., 2012) and many other publications, however, also propose other mechanisms for the formation of flowstripes. The most prominent to consider here is the formation of flowstripes due to glacier confluence which is clearly the case here. How can you exclude other mechanisms than bed bumps for the formation of flowstripes? If this can't be done, then linking the bedforms between the two radar transects is maybe not that straightforward. *The referee is correct to point to glacier confluence as a cause for flowstripes. The same processes forming the surface features will affect the bed, however. That said, as ice accelerates as a consequence of water present at the bed and enhanced longitudinal stresses, erosion at the bed will occur along the direction of ice flow. To say otherwise would be a major objection to a large element of glacier geological work. We really don't see any controversy here.*

I suggest an additional Figure marking the different flowstripes and their origin (e.g. Fig. 9 in Ely et al., JGR, 2017). If 4 flowstripes can be uniquely attributed to the basal landforms 1-4, then this would strengthen the paper. Do the flowstripes have a characteristic end-point somewhere farther upstream, potentially marking the alongflow extent of the basal landforms? *We use the flowstripes simply to work out the ice flow direction. The bedforms do map onto this direction, as one would expect from glacial geological interpretation. And does the simple reflectivity analysis, highlighting where basal water is likely. Given this association – not co-genesis – we are able to reliably infer water and bed forms along the line of ice flow. However, on close analysis of the enlarged map we see that the flowstripes terminate at the flexure grounding line, meaning that they are preserved over the ‘estuary’ section of the grounding zone.*

Radar reflectivity analysis: I appreciate the interpretation of the radar amplitudes for the detection of potentially localized basal water. However, the point that needs to be made here is that areas of high basal reflectivity correspond with the predicted water flow paths. Therefore, I suggest to directly mark the three subglacial water channels (and their width, and preferably uncertainty) directly in the corresponding radargrams. The hydrostatic potential is in my eyes in this regard not very useful. *We have done this, as requested. We agree that the hydrostatic potential is less important and have removed it.*

Comparison with previous findings: In l. 242 briefly compares the new findings to our previously published findings (Drews et al., 2016, Nat. Comm.) and suggests that ‘our’ features could also be interpreted as hard-bedded features. I am not against this interpretation and, in hindsight, I surely should have considered this hypothesis in more detail. *Of course many scientific developments are a consequences of refinement, and this may be one example. The fact is that the literature is presently bereft of analysis of hard-rock glacial geology influencing ice flow – aside from palaeo work. We believe hard-rock geology has been overlooked as a modulator of ice and water flow – and indeed we have now seen similar bed roughness upstream of other Weddell sector ice shelf channels (upcoming work, which we would be happy to share here).*

What I am missing is a more in-depth comparison of the different datasets and their interpretations. I am currently unsure if we (i) see the same feature and have different interpretations, (ii) see different features, or (iii) if we focus on different aspects of the same features (i.e. hard-bed landforms which are covered by sediments). Here are some thoughts from my side about this issue, and it may be informative to explore some of these aspects:

1. Figure 1 in our paper shows that Reflectors A-C have much internal structure (I don’t think those are off-angle reflectors). This is inconsistent to me with a hard-bed landform. (Maybe we are observing different features?). *That is possible, but we would caution against using radar to infer internal structure. We don’t believe it can give unequivocal results.*
2. We see some evidence, that the bumps in the surface topography linked to the basal landforms are (sometimes) advected into the adjacent ice shelves (Fig. 7). We have interpreted this as sign for “activity” of the basal landform which is inconsistent with a hard-bed feature. (Maybe we are observing different features?) *We do not see evidence of bedform advection – it is possible, of course, that there are different systems being viewed.*
3. Our landforms are max. 300 m wide, which is 3-13 times narrower than what is observed in this paper. (Maybe we are observing different features?). *Again, yes this is possible. The landforms seem in Foundation ice stream are enormous – one of the striking elements of the work, and they are simply too large to be sediment forms.*
4. Our landforms are not linked by flowstripes, and are not in a setting of glacier confluence. (Maybe we are observing different features?) *I think the flowstripe issue is a red herring. The only use of them is to determine easily ice flow direction, not causality.*

5. We think (but have no hard evidence) that “our” landforms only reach their large height in a narrow (maybe 1-2 km) region very close to the grounding line. This is inconsistent with your observations which show nicely that the landforms extend much farther upstream. (Maybe we are observing different features? Maybe our interpretation is wrong?) *This would be difficult to say without further work, in order for you to be more certain of course.*

On reflection, we did not intend to claim that other analysis was wrong. Just that hard-bedded explanation had not previously been made, and that it represents a plausible (and we think widespread, given the glacial geological landforms in previously glaciated regions, including offshore Antarctica, as we discuss in SI) explanation for ice shelf channels. Our main concern with the sediment eskers explanation is that the sheer size does not have an obvious glacial geological comparator. The features we observe are too large to realistically be interpreted as ‘eskers’. Also, eskers are generally sinuous – we do not observe any sinuosity in the bedforms beneath Foundation Ice Stream.

Small Remarks

l. 32: “that”  “than” (?) **corrected.**

l. 123 “They are divided in distinct sets relative to the origin of the ice”. I think the following lines only describe the ridges on the Academy Glacier. Not sure where the other “distinct set” vis taken up again. **We mean the separation between East and West Antarctic origin ice.**

l. 233 I am lost what confirms the “zig-zag” of the grounding line, and how this relates to the large tidal uplift of 18.5 km. Does the “zig-zag” have clear correspondence in the surface topography? **The zig-zag grounding line is from the slopes analysis, and is as expected from a highly corrugated bed – though that has never been pointed out.**

l. 434 “or” – “of” **changed**

l. 434 is it not “four” (instead of “three”) channels and landforms ? **changed**

I like the terminology of using M and U channels and will use it in the future. I also very much like Supplementary Figure S4 adding good evidence for the hard-bed landform interpretation. Thanks. We have moved S4 into the main text following comments from Referee #2.

Figure 1: Academy Glacier, Support Force Glacier and Foundation Ice Stream are not labelled in the Figures. **This has been changed.**

Figure 1 does not convey the merging of M-channels. It is simply not visible, especially in print. **We have contrast stretched the MODIS data to make it clearer. We’ve also added some arrows to show where the M-channels are.**

Figure 1d should additionally mark the locations of the landforms+the calculated water flowpaths, otherwise, how should the reader make the connection that these landforms guide the water flow? I also suggest to zoom much more into the survey area, I don’t think it is needed to always show the entire catchment. **We have now done this, as requested.**

Figure 2 Profiles M-M’ and N-N’ mark the trunk area referred to in l. 111. **OK**

Figure 2 Mark flow direction of ice (into our out-of-plane). Mark surface and multiples. **OK**

Figure 2 Indicate clearly if I-I’ is grounded or not. **Done**

Figure 2: How do you define the height of “mega-ridge” 2 in profile K-K’? There is different heights depending if you choose left or right side no? **We choose the maximum value.**

Figure 2: It is probably better to plot relative to sea level not WGS84. Double check the 0 in LL’? This makes no sense to me. We commonly choose WGS84 – it doesn’t make a great deal of difference and doesn’t affect the analysis. **No changes made.**

Figure 2: It would be helpful to have a subplot in each transect showing the surface topography in a blown-up version. It would be nice to know if the basal bedforms are accompanied by surface bumps (marking the flowstripes?), and where the M-channels are in relation to the U channels. If such

topography is not available state that explicitly as a limitation. **We have put surface altimetry above all of the radargrams.**

Figure 2: Profile L – L' left side, I don't see four U Channels. Only three. (What is arrow 4 pointing to?). **We have limited our analysis to three channels.**

Figure 3: Width and Height y-axis should start at 0. Mark position (or better range) of grounding-line estimates in these plots as well. (i. e. k-j grounded, h floating). **Changed.**

Figure S1: Don't you use an updated version of Bedmap 2? Caption does not state this. Not sure what this Figure adds to the entire story. I would prefer a zoom to the FIS area (and some measure of uncertainty as mentioned above). **This has been done, as requested.**

Figure S2. There are no blue, orange or green lines indicating the different areas of the grounding zone. There is only one yellow line. **These are dots not lines – the figure caption now makes this clearer.**

REVIEWERS' COMMENTS:

Reviewer #1 (Remarks to the Author):

The revised version of this paper is very much improved, and the detailed & kind rebuttal has cleared up most of my initial reservations. Although I do not agree with all points in the rebuttal, the main message of the paper is now much clearer, and secondary points (e.g. meandering/merging, the subglacial hydrological modelling) have been cleared up or removed. Below, I mention a number of minor points which can be addressed quite easily. I thank the authors for taking the review comments seriously, and compliment them on a study well done.

- (1) I suggest to add two back of the envelope calculations (detailed below) which would quantify the required basal melt rates (for accentuating the corrugations), and for the timing of the ice-shelf channel formation (based on channel length and advection time). Those numbers could be useful for other studies.
- (2) It should be stated more clearly, that the corrugation alone (by basal landforms higher than 100 m) will also cause formation of M-channels. The M-channels will be certainly smaller without the plume melting, but I think they should still be visible. Figure S7 should be adapted accordingly, and explanation for U1 rephrased. I also wonder if Figure S7 should be moved to the main manuscript. Showing it early on would introduce the main idea of the paper and clarify the terms U/M channels which may be otherwise confusing for readers outside glaciology.
- (3) I am not yet satisfied by the author's response to my point 4 regarding the meandering and merging of ice-shelf channels. The authors clearly "observe" the merging/meandering, but they still do not "explain" it. On the contrary, the elongated landforms pinpoint the water outlet at the grounding line, therefore ice-shelf channels should be along flow lines (as illustrated in Figure S7). Any meandering/merging that is seen farther seawards is thus more likely a consequence of oceanographic processes occurring farther away from the grounding line, which is not further explored in the paper. (The alternative hypothesis, that the subglacial water outlet positing is variable over time is now less likely, because the outlet is pinned by the hard-bed landform). Although the paper has made progress on this point, I wouldn't yet call it an explanation. **However, because the corresponding statements with respect to merging/meandering have been removed in the revised version, I have no problem with this point anymore.**

Minor edits:

L 49: Reference [,3] should be [3] (or a reference is missing)

L 100: Can you tell if the downstream ends of the 150 km M-channels are advecting with ice flow? If so, you could calculate in a back-of-the envelope calculation at which time the channel has initially formed at the grounding line. Such information could be useful for other studies investigating past ice-dynamic changes linked to the subglacial hydrology. If the downstream ends are not advecting, this would also be an important finding (all cases I have seen so far are advecting)

Profile N-N'/M-M': Mark the years of the data on which the MOA, the DinSAR, and the radar GL are based. This suggests more clearly that the GL may be transient (rather than a misinterpretation of the underlying data)

I. 140: State here how far you can trace these landforms inland (giving a lower limit on their longitudinal extent)

I. 152: The esker ramps (or tadpole shaped eskers) inferred by our previous study only reach their large height very close to the grounding line (i.e. 1-2 km). Elsewhere they should be much smaller. So instead of invoking the missing sinuosity (which may be undersampled by the profile spacing), I would rather state that the ridges observed here are very high even tens of kilometers upstream of the grounding line. This is much better explained with a hard-bedded landform (and inconsistent with tadpole shaped esker).

I. 166 "ie." -> "i.e."

I. 190 "bed reflection" -> "basal reflection" ; bed reflection reminds me on ice-bed interface, but here it is more an ice-ocean interface.

I. 190: A back of the envelope calculation would be useful here: if profile J-J' is a grounded rock pinnacle, and I-I' is a floating U-channel. How much melting is required to explain the height difference between the two (using today's surface velocities)? This would put a (lower limit) number on the accentuation of the channels by plume melting.

I. 229 Channel U1 has no connection to the grounding line. Therefore, I think it is unlikely that this channel can also be linked to a 100 m high, hard-bedded landform upstream of the GL. If there was such a landform, I would still expect a 100 m corrugation farther seawards, which would correspond to roughly a 10 m surface depression (which should be visible in MODIS imagery). It could be due to transient changes in subglacial hydrology (i.e. a water outlet that is not pinned by a hard bedded landform), or it could be pure ocean forcing. Consider rephrasing this paragraph. Also adapt Figure S7 accordingly (see comment below).

Caption Figure S3: "Isurface" -> "(surface "
Figure S3a y labels should be "elevation", not "height".

Figure S7: The corrugation by hard-bedded landforms of this size (> 100 m) will also cause an M-channel farther seawards, independent of the subglacial hydrology. This M-channel will be smaller, but should still be visible in satellite imagery. I suggest to adapt the Figure accordingly.

Warning: Nat. Comms. does not accept “-“ in the title. After a number of revisions during typesetting I gave up and renamed “ice-shelf channel” to “ice shelf channel”. In case this paper will be accepted, I hope you will have more perseverance than me (same for hard-bed)

[Redacted]

Editorial Note: Parts of this Peer Review File have been redacted as indicated to maintain confidentiality.

Reviewer #2 (Remarks to the Author):

I have read the several reviews of the manuscript and re-examined the text and figures. I think the authors have done an excellent job of repoyndig to the review comments, defending their ideas and adjusting text and especially figures where requested. The reviews were alternately helpful with good suggestions and stubborn due largely to the challenges of crossing disciplines, even when the disciplines are a closely related as ice sheet glaciology and glacial geology.

The case is made much clearer now that the authors are showing that grooves incised from bedrock landforms initiate some flowstripes, and further, that some of these grooves are further worked by either sub-glacial outflow of water or tidally-pumped and/or density-driven ocean circulation.

The reveiwers instinctively wanted frugality in the number of figures, but the authors are correct that the online format of Nature Communications obviates the need for limited figures if they are of value. As evidence of their conclusions, and as an easily-accessible record of what data was collected, the figures are valuable. The revised text has improved them significantly.

Reviewer #3 (Remarks to the Author):

A much better paper with now excellent illustrations.

Details of changes made

Referee #1 made some important final suggestions to improve the paper, which we deal with below (in red). Many of the changes made are included in the newly formed Discussion section.

Referee #1.

The revised version of this paper is very much improved, and the detailed & kind rebuttal has cleared up most of my initial reservations. Although I do not agree with all points in the rebuttal, the main message of the paper is now much clearer, and secondary points (e.g. meandering/merging, the subglacial hydrological modelling) have been cleared up or removed. Below, I mention a number of minor points which can be addressed quite easily. I thank the authors for taking the review comments seriously, and compliment them on a study well done.

We thank the referee for appreciating the changes made, and the value of our work.

(1) I suggest to add two back of the envelope calculations (detailed below) which would quantify the required basal melt rates (for accentuating the corrugations), and for the timing of the ice-shelf channel formation (based on channel length and advection time). Those numbers could be useful for other studies.

This is an interesting suggestion. We are cautious about this approach, because the influence of tides, and the amount of advection is not known well. This is certainly an area for future modelling work, and we now allude to this in the newly formed Discussion section.

(2) It should be stated more clearly, that the corrugation alone (by basal landforms higher than 100 m) will also cause formation of M-channels. The M-channels will be certainly smaller without the plume melting, but I think they should still be visible. Figure S7 should be adapted accordingly, and explanation for U1 rephrased. I also wonder if Figure S7 should be moved to the main manuscript. Showing it early on would introduce the main idea of the paper and clarify the terms U/M channels which may be otherwise confusing for readers outside glaciology.

Again, another interesting point. Without a supply of water to melt the channel (from the subglacial system or from tides) we expect the channels to close due to creep of the ice, downstream of the landforms. We now make this point clearer in the Discussion section.

(3) I am not yet satisfied by the author's response to my point 4 regarding the meandering and merging of ice-shelf channels. The authors clearly "observe" the merging/meandering, but they still do not "explain" it. On the contrary, the elongated landforms pinpoint the water outlet at the grounding line, therefore ice-shelf channels should be along flow lines (as illustrated in Figure S7). Any meandering/merging that is seen farther seawards is thus more likely a consequence of oceanographic processes occurring farther away from the grounding line, which is not further explored in the paper. (The alternative hypothesis, that the subglacial water outlet positioning is variable over time is now less likely, because the outlet is pinned by the hard-bed landform). Although the paper has made progress on this point, I wouldn't yet call it an explanation. **However, because the corresponding statements with respect to merging/meandering have been removed in the revised version, I have no problem with this point anymore.**

The referee is right that we 'observe' but do not 'explain' the meandering of the channels in the ice shelf. We now discuss the issue in the Discussion section, and put forward the most likely option.

Minor edits:

L 49: Reference [3] should be [3] (or a reference is missing) **changed**.

L 100: Can you tell if the downstream ends of the 150 km M---channels are advecting with ice flow? If so, you could calculate in a back---of---the envelope calculation at which time the channel has initially formed at the grounding line. Such information could be useful for other studies investigating past ice-dynamic changes linked to the subglacial hydrology. If the downstream ends are not advecting, this would also be an important finding (all cases I have seen so far are advecting) Profile N---N'/M---M': Mark the years of the data on which the MOA, the DinSAR, and the radar GL are based. This suggests more clearly that the GL may be transient (rather than a misinterpretation of the underlying data).

We are unable to answer whether the M channel at the seaward end is advecting. Surely advection will be occurring, but so too will channel development through subglacial water input and tidal influences. It is likely, seeing that the tidal range is so high and the channels exist so far from the grounding line, that tidal influences are important once a channel has established, and we now refer to this idea in the Discussion section.

I. 140: State here how far you can trace these landforms inland (giving a lower limit on their longitudinal extent).

We are able to trace the landforms ~40 km inland of the grounding zone – we now make this clear.

I. 152: The esker ramps (or tadpole shaped eskers) inferred by our previous study only reach their large height very close to the grounding line (i.e. 1-2 km). Elsewhere they should be much smaller. So instead of invoking the missing sinuosity (which may be undersampled by the profile spacing), I would rather state that the ridges observed here are very high even tens of kilometres upstream of the grounding line. This is much better explained with a hard-bedded landform (and inconsistent with tadpole shaped esker).

Good point. We have added that explanation to the paper.

I. 166 "ie." - "i.e." **changed**.

I. 190 "bed reflection" - "basal reflection" ; bed reflection reminds me of ice-bed interface, but here it is more an ice---ocean interface. **Changed**.

I. 190: A back of the envelope calculation would be useful here: if profile J-J' is a grounded rock pinnacle, and I-I' is a floating U---channel. How much melting is required to explain the height difference between the two (using today's surface velocities)? This would put a (lower limit) number on the accentuation of the channels by plume melting.

As noted above, this would be difficult to do reasonably without some further work. It appears simple, but actually the level of advection vs tidal influences are important yet poorly known at the moment. We discuss this issue in the Discussion section.

I. 229 Channel U1 has no connection to the grounding line. Therefore, I think it is unlikely that this channel can also be linked to a 100 m high, hard-bedded landform upstream of the GL. If there was such a landform, I would still expect a 100 m corrugation farther seawards, which would correspond to roughly a 10 m surface depression (which should be visible in MODIS imagery). It could be due to

transient changes in subglacial hydrology (i.e. a water outlet that is not pinned by a hard bedded landform), or it could be pure ocean forcing. Consider rephrasing this paragraph. Also adapt Figure S7 accordingly (see comment below).

We agree that the switch-off in basal hydrology here can explain why the M-channel is not connected to the grounding line – being advected downstream, yet also maintained by tidal pumping. We discuss this point in the Discussion section.

Caption Figure S3: “Isurface” - “(surface“ **changed**

Figure S3a y labels should be “elevation”, not “height”. **Changed.**

Figure S7: The corrugation by hard---bedded landforms of this Size (> 100 m) will also cause an M-channel farther seawards, independent of the subglacial hydrology. This M---channel will be smaller, but should still be visible in satellite imagery. I suggest to adapt the Figure accordingly.

We’re not sure it would. The ice may likely experience ‘creep closure’ of the channel without water to melt and maintain it. Hence, we do not make any changes to Supplementary Figure 7. We also wish to keep this figure in Supplementary Information, rather than the main paper – although as may use it in our publicity of the work.